# Suppression of ACE2 SUMOylation protects against SARS-CoV-2 infection through TOLLIP-mediated selective autophagy

Shouheng Jin [1,5] ✉, Xing He[1,5], Ling Ma[1], Zhen Zhuang[2], Yiliang Wang[2], Meng Lin[1], Sihui Cai[1], Lu Wei[1], Zheyu Wang[1], Zhiyao Zhao[2], Yaoxing Wu[1], Lin Sun[3], Chunwei Li [3], Weihong Xie[1], Yong Zhao[1], Zhou Songyang [1], Ke Peng [4], Jincun Zhao [2] & Jun Cui [1] ✉

In addition to investigating the virology of severe acute respiratory syndrome coronavirus 2 (SARS-CoV-2), discovering the host–virus dependencies are essential to identify and design effective antiviral therapy strategy. Here, we report that the SARS-CoV-2 entry receptor, ACE2, conjugates with small ubiquitin-like modifier 3 (SUMO3) and provide evidence indicating that prevention of ACE2 SUMOylation can block SARS-CoV-2 infection. E3 SUMO ligase PIAS4 prompts the SUMOylation and stabilization of ACE2, whereas deSUMOylation enzyme SENP3 reverses this process. Conjugation of SUMO3 with ACE2 at lysine (K) 187 hampers the K48-linked ubiquitination of ACE2, thus suppressing its subsequent cargo receptor TOLLIP-dependent autophagic degradation. *TOLLIP* deficiency results in the stabilization of ACE2 and elevated SARS-CoV-2 infection. In conclusion, our findings suggest selective autophagic degradation of ACE2 orchestrated by SUMOylation and ubiquitination as a potential way to combat SARS-CoV-2 infection.

The ongoing coronavirus disease 2019 (COVID-19) pandemic due to severe acute respiratory syndrome coronavirus 2 (SARS-CoV-2) infection has posed an extraordinary threat to global health and economy[1,2]. Coronaviruses are diversified enveloped viruses that can infect many different vertebrates and cause mild to severe respiratory and intestinal infections in humans[3]. The trimeric coronavirus spike (S) glycoproteins interact with their matching host receptors for infection. During the viral entry of SARS-CoV-2, the S protein of SARS-CoV-2 (SARS-2-S) is cleaved into the S1 and S2 subunits, which is essential for the recognition and membrane fusion to host cells. Angiotensin-converting enzyme 2 (ACE2) serves as a host cell target receptor of SARS-CoV[4,5]. Owing to the high similarity between the spikes of SARS-

CoV and SARS-CoV-2, researchers have uncovered that ACE2 is the dominated entry receptor of SARS-CoV-2 and SARS-CoV-2 binds ACE2 through the C-terminal domain (also named as the receptor-binding domain (RBD))[6,7]. Therefore, the studies focus on peptide, antibody and small chemical compound targeting ACE2 can be applied to the treatment of COVID-19[8–10]. However, the precise regulation of ACE2 upon SARS-CoV-2 infection remains largely unclear.

Autophagy is a conserved eukaryotic degradation system to sequester intracellular components into autophagosomes, which follows by fusing with lysosomes to degrade the captured substrates for recycle[11,12]. Emerging evidence indicates that substrates are engulfed into autophagosomes in a selective manner, which is mediated by

[1]Guangdong Province Key Laboratory of Pharmaceutical Functional Genes, MOE Key Laboratory of Gene Function and Regulation, State Key Laboratory of Biocontrol, School of Life Sciences, Sun Yat-sen University, 510275 Guangzhou, Guangdong, China. [2]State Key Laboratory of Respiratory Disease, Guangzhou Institute of Respiratory Disease, The First Affiliated Hospital of Guangzhou Medical University, 510182 Guangzhou, Guangdong, China. [3]Department of Otolaryngology, First Affiliated Hospital, Sun Yat-sen University, 510080 Guangzhou, Guangdong, China. [4]State Key Laboratory of Virology, CAS Key Laboratory of Special Pathogens, Center for Biosafety Mega-Science, Wuhan Institute of Virology, Chinese Academy of Sciences, 430071 Wuhan, Hubei, China. [5]These authors contributed equally: Shouheng Jin, Xing He. ✉e-mail: jinshh3@mail.sysu.edu.cn; cuij5@mail.sysu.edu.cn

autophagy cargo receptors, such as p62/SQSTM1, nuclear dot protein 52 (NDP52), NBR1 (neighbor of BRCA1), and Toll interacting protein (TOLLIP)[13,14]. These receptor proteins tackle ubiquitinated substrates through their ubiquitin-binding domains and are linked with autophagosomes via the light chain 3 (LC3)-interacting regions[15]. So far, which linkage types of polyubiquitin chains can target to special cargo for autophagic degradation are yet to be fully characterized[16], and understanding of the underlying mechanism of selective autophagy prompts developments of autophagy-modulating medicine with cargo selectivity[17].

Reversible post-translational modification mediated by small ubiquitin-like modifier (SUMO) participates in a highly dynamic process of cellular functions, including protein localization, interaction, stabilization, as well as the activity of substrates[18,19]. In mammals, SUMOylation entails an enzymatic cascade mediated by the E1 SUMO-activating enzyme (SAE), the E2-conjugating enzyme (UBC9), and a limited group of E3 SUMO ligases[20]. Conversely, SUMO/Sentrin specific proteases (SENPs) quickly cleave the SUMO to reverse this process[21].

In this study, we found E3 SUMO ligase PIAS4 promotes the conjugation of SUMO3 with ACE2, while deSUMOylating enzyme SENP3 decreases the SUMOylation level of ACE2. The covalent association of SUMOylation of ACE2 dampens the K48-linked ubiquitination of ACE2 at lysine (K) 187, thereby hindering TOLLIP-ACE2 interaction along with the degradation of ACE2 through selective autophagy. Considering the importance of ACE2 in the viral entry during early stages of SARS-CoV-2 infection, we demonstrate that loss of SUMOylation reduces viral infection by mediating ACE2 destabilization through autophagic degradation. Our findings illustrate that SUMOylation restrains the degradation of ACE2 receptor through TOLLIP-mediated selective autophagy to increase the host susceptibility for SARS-CoV-2, which might be a potential target for COVID-19 therapy.

## Results

### ACE2 undergoes SUMO modification

To delineate the large-scale proteomic landscape of ACE2 interaction and regulation, we adopted mass spectrometry (MS) using ACE2 as a bait and identified 773 ACE2-ineracting proteins (Supplementary Data 1), among which we discovered a portion associated with SUMOylation by means of Gene Ontology (GO) enrichment analysis towards biological process, one of three major GO analysis terms (Fig. 1a). Protein conjugation with SUMO (Fig. 1b), has attracted considerable attention owing to its indispensable roles in mammalian cells[21]. We immunoprecipitated SUMO from cellular lysates under denaturing conditions to avoid the isolation of large protein complexes, and detected high apparent band of ACE2 in the SUMO2/3/4 immunoprecipitates (Fig. 1c). SUMO2, SUMO3 and SUMO4 are virtually identical[18], and there is no commercial antibody to be capable of distinguishing them. To figure out which type of SUMO could be conjugated with ACE2, we next co-overexpressed different types of SUMO with ACE2 and observed that ACE2 specifically linked with SUMO3. Moreover, the amount of SUMOylated ACE2 was increased following UBC9 overexpression (Fig. 1d). Due to the C-terminal di-Gly motif of SUMO3 in the attachment to substrates, the conjugation of ACE2 was totally abrogated within the mutant form of SUMO3 (SUMO3 GG/AA) (Fig. 1e). Furthermore, we observed that cells treated with ML-792, the selective inhibitor of SAE/SUMO (Fig. 1f, the respective target of compound can be found in 2. Supplementary Table 1), exhibited dramatically lower colocalization of SUMO2/3 and ACE2 (Fig. 1g and Supplementary Fig. 1a, b). The coexistence of ACE2 and SUMO2/3 lead to the localization of SUMOylated ACE2 at the plasma membrane (Supplementary Fig. 1c). Bioinformatics analysis of ACE2 predicted five potential SUMO-conjugation consensus motifs (Fig. 1h and Supplementary Fig. 1d, e). We constructed ACE2 mutants bearing lysine (K) to arginine (R) substitution in the predicted SUMOylation site for in vivo

SUMOylation assay, observing that only K187R ACE2 mutant displayed decreased SUMOylation (Fig. 1i). We next found that SARS-CoV-2 infection reduced the SUMOylation levels of wild-type (WT) ACE2, but not the K187R mutant form of ACE2 (Fig. 1j), indicating that K187 is a pivotal SUMOylation site of ACE2.

### Impaired ACE2 SUMOylation suppresses SARS-CoV-2 infection

Since ACE2 is the major recognition receptor for the envelope coated by SARS-2-S, we examined the function of ACE2 SUMOylation in SARS-CoV-2 infection. We generated SARS-CoV-2 S pseudotyped virus containing the spike protein, but not any nucleic acid of SARS-CoV-2, so it cannot replicate in host cells, which serves as a good model for the study of viral entry (Fig. 2a). We observed that the SUMOylation of ACE2 and the cellular pseudotyped SARS-CoV-2 levels were largely decreased with the treatment of SUMOylation inhibitors (Fig. 2b, c). We assessed the cytotoxicity of SUMOylation inhibitors using a lactate dehydrogenase (LDH) assay and found that only the high concentration of GA (100 μM) and TAK1-981 (50–100 μM) had slight influence on the viability of Calu-3 cells (Supplementary Fig. 2a). We treated human pulmonary alveolar epithelial cells (AECs) with SUMOylation inhibitors, finding that the SUMOylation of ACE2 and infection of pseudotyped SARS-CoV-2 were largely abrogated by SUMOylation inhibitors (Fig. 2d, e). To investigate the in vivo protection roles of SUMOylation inhibitors in SARS-CoV-2 infection, 6-week-old BALB/c mice were transfected intranasally with $2.5 \times 10^8$ plaque forming unit (PFU) of adenovirus expressing human ACE2 (Ad5-hACE2) as previously described[22]. Mice received indicated SUMOylation inhibitor treatment for 2 days were challenged with SARS-CoV-2 ($1 \times 10^5$ PFU of) for 3 days (Fig. 2f). Our results showed that both ML-792 and GA treatment shut down the infection of SARS-CoV-2 (Fig. 2g and Supplementary Fig. 2b).

To provide the insights into SARS-CoV-2 infection, we next carried out RNA-sequencing (RNA-seq) analysis in Calu-3 cells infected with SARS-CoV-2 for 24 h, together with SUMOylation inhibitor 2-D08 treatment. The cutoff of fold-change (FC) ratio ($|\log_2 FC| \geq 1$) and P-value (p-value < 0.05) were employed to identify differentially expressed genes (DEGs). Gene set enrichment analysis (GSEA) showed downregulation of genes associated with 'viral genome replication' (Fig. 2h), suggesting that 2-D08 treatment provided protection against SARS-CoV-2. The activation of immune pathways, as well as the transcription of cytokines and chemokines upon SARS-CoV-2 infection were decreased by 2-D08 treatment (Fig. 2i, j). Genes associated signatures with SARS-CoV-2 challenge (including *IFNB1*, *ISG15*, *IFNL1*, *IFNL2*, *CXCL10*, and *CCR1*) were identified in IGV genome browser (Supplementary Fig. 2c). Additional qPCR analysis further confirmed the inhibitory roles of 2-D08, ML-792, and GA in SARS-CoV-2 infection (Supplementary Fig. 2d–f). Altogether, these results demonstrated that suppressing global SUMOylation affects ACE2 SUMOylation and inhibits the infection of SARS-CoV-2.

### SUMO3 conjugation promotes the stabilization of ACE2

We evaluated the ACE2 turnover and observed that the protein levels but not the mRNA levels of ACE2 were gradually reduced upon ML-792 and 2-D08 treatments in a dose-dependent manner (Fig. 3a, b). Calu-3 cells treated with GA and TAK-981, also showed diminished ACE2 protein abundance (Supplementary Fig. 3a). Additionally, treatments of both ML-792 and 2-D08 attenuated the ACE2 abundances in human nasal epithelial progenitor cells (hNEPCs, a type of human primary cells), human umbilical vein endothelial cells (hUVECs) and HepG2 cells (Supplementary Fig. 3b–d). Moreover, 2-D08-induced ACE2 degradation was almost abolished when the K187 of ACE2 is mutated (Fig. 3c). To better understand the SUMOylation-mediated ACE2 stabilization in the initial attachment of SARS-CoV-2, we studied the roles of deSUMOylation enzymes under the infection of SARS-CoV-2 S pseudotyped virus. Firstly, we transduced ACE2 constitutively expressing A549 cells with an all-in-one lentiviral vector containing

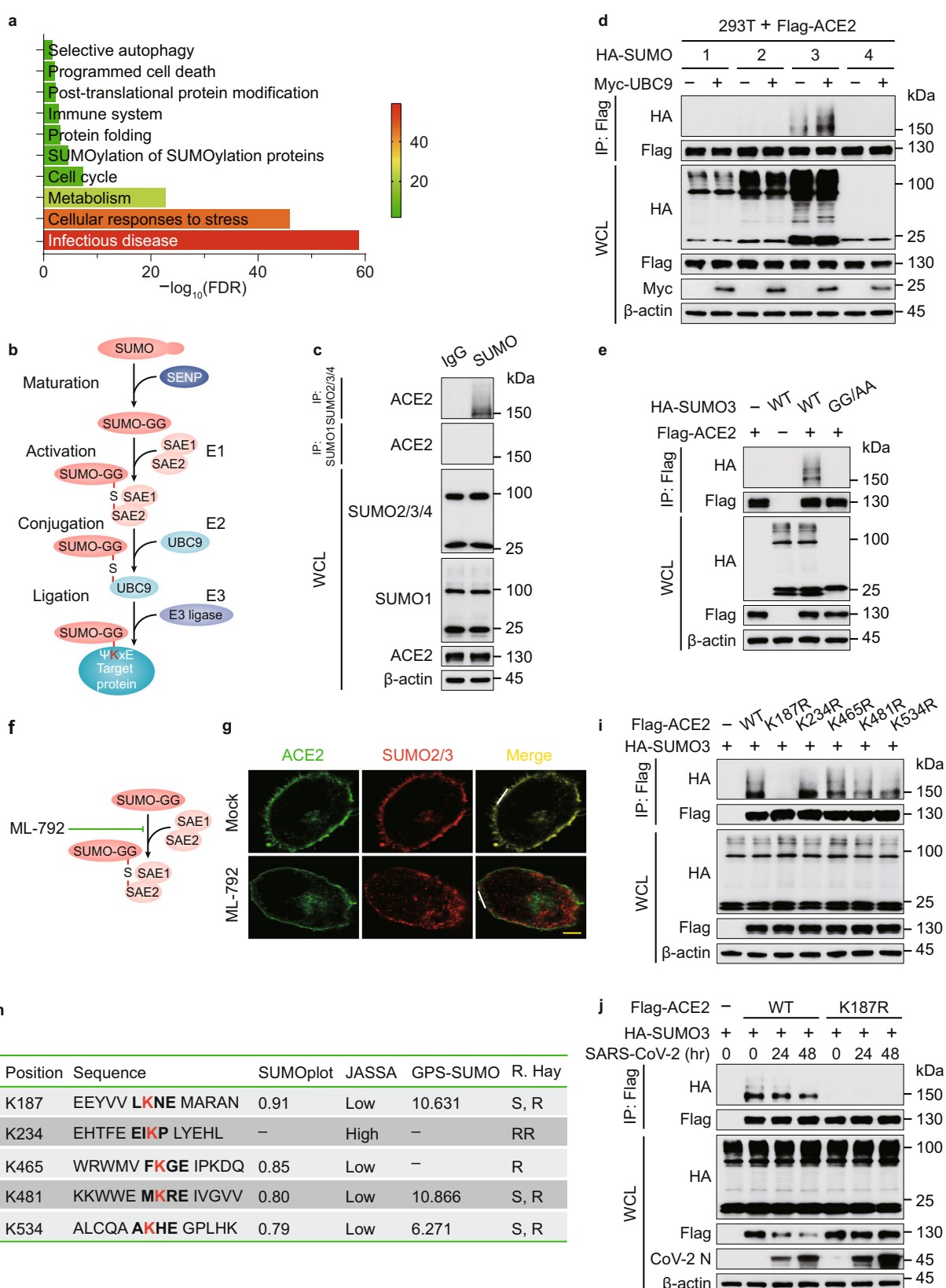

small guide RNA (sgRNA) from 7 SENP deSUMOylating enzymes, Cas9, and a puromycin resistance gene. When puromycin selection was completed, the selected cells were cultured for 9 days to ensure SENPs depletion. We next treated the cells with SARS-CoV-2 S pseudotyped virus (Fig. 3d), finding that *SENP3* depletion dramatically enhanced the infection of SARS-CoV-2 S pseudotyped virus by luciferase assay

(Fig. 3e). We applied specific siRNAs targeting *SENP3* and obtained similar results (Fig. 3f). These data implied SENP3 works as a suppressive regulator for SARS-CoV-2 infection.

We next verified the interaction between SENP3 and ACE2 (Supplementary Fig. 3e). SENP3 overexpression promoted the ACE2 degradation (Supplementary Fig. 3f), whereas the protein abundances

**Fig. 1 | ACE2 receptor is specifically conjugated with SUMO3. a** Lysates prepared from HEK293T cells transfected with vector encoding Flag-tagged ACE2 or Flag-tagged empty vector as negative control were immunoprecipitated with α-FLAG M2 beads. The precipitated proteins were subjected to mass spectrometry analysis. This experiment was performed once with two replicates per condition. Gene Ontology (GO) analysis of the proteome after purified with ACE2 receptor as bait. The color code displays the value of (−log$_{10}$FDR) of the respective GO terms. **b** Schematic diagram of the conjugation of small ubiquitin-like modifier (SUMO) with target proteins. **c** Extracts of Calu-3 cells were immunoprecipitated with anti-SUMO and immunoblotted with indicated antibodies. **d** Lysates of 293 T cells transfected with plasmids of Flag-ACE2 and HA-tagged SUMO, together with Myc-UBC9 were immunoprecipitated with α-FLAG M2 beads and immunoblotted with anti-HA. **e** Immunoprecipitation and immunoblot analysis of 293T cells transfected with expressing vectors of Flag-ACE2 and HA-SUMO3 and its indicated mutants. **f** Illustration of the inhibitory site of ML-792. **g** HeLa cells transfected with Flag-

ACE2 in the absence or presence of ML-792 (10 μM), followed by labeling of ACE2 (green) and SUMO2/3 (red) with specific antibodies. Scale bar, 20 μm. A representative experiment out of three is shown. **h** Schematic representation of the five predicted SUMO-conjugation motifs (ψKXE) in ACE2 protein. Details of the SUMO-conjugation motifs and summary of prediction software scores are depicted underneath. Analysis using Ron Hay's website (https://www.lifesci.dundee.ac.uk/groups/ron_hay/pages/SumomotifQuery.html) provides motif types: S = strict, R = relaxed, and RR = relaxed reverted. **i** Immunoprecipitation and immunoblot analysis of 293 T cells transfected with vectors encoding HA-SUMO3 and wild-type (WT) Flag-ACE2 or its indicated mutants. **j** 293 T cells transfected with vectors expressing Flag-ACE2 (WT or K187R) and HA-SUMO3 along with SARS-CoV-2 (MOI = 0.5) infection for indicated time points, followed by immunoprecipitated with α-FLAG M2 beads and immunoblot analysis. For **c–e, i, j**, similar results are obtained by three independent biological experiments. Source data are provided as a Source data file.

of ACE2 and the infection of SARS-CoV-2 were increased by *SENP3* deficiency (Fig. 3g). The SUMO3 modification of WT ACE2, but not ACE2 K187R mutant, was elevated by *SENP3* depletion (Fig. 3h), indicating that SENP3 functions as a deSUMOylating enzyme to cleave the attachment of SUMO3 on ACE2 at K187 site. To determine which SUMO E3 ligase is responsible for ACE2 SUMOylation, we observed that PIAS4 interacted with ACE2 (Supplementary Fig. 3g, h), and the protein amount of ACE2 positively correlated with PIAS4 (Supplementary Fig. 3i). To investigate the function of ACE2 SUMOylation in SARS-CoV-2 infection, we further observed that *PIAS4* depletion downregulated ACE2 stabilization and suppressed SARS-CoV-2 infection (Fig. 3i, j). PIAS4 lost its ability to promote the SUMOylation and stabilization of K187R ACE2 (Fig. 3k). Altogether, these findings indicated that dynamic SUMOylation of ACE2 at K187 promotes its stabilization, thereby influencing the SARS-CoV-2 infection.

We next detected whether ACE2 SUMOylation is dependent on the ER/Golgi pathway, as ACE2 contains a putative N terminal signal peptide (SP, aa 1–17)[23]. We enriched the integral membrane proteins and membrane-associated proteins from cultured Calu-3 cells, observing that ACE2 could be displayed both in the membrane fraction and soluble cytosolic fraction. Next, we carried out co-immunoprecipitation assay and found that PIAS4/SENP3/TOLLIP co-existed with ACE2 in the soluble cytosolic fraction (Supplementary Fig. 3j). These results suggested that ACE2 SUMOylation mainly occurs in the soluble cytosol. Interestingly, ACE2 is recently to be reported to distribute in the cytomembrane, soluble cytoplasm as well as nucleus, and the aa 1–740 of ACE2 undergoes multilayer post-translation modifications by AMPK, Casein kinase 1 alpha (CK1α), and LSD1[24–26]. Moreover, the ACE2 ΔSP protein could still localize on the cell membrane (Supplementary Fig. 3k). Our results suggested that besides the ER/Golgi pathway, there are other possible mechanisms that modulate the plasma membrane localization and cellular shuttle of ACE2. To better understand the cellular trafficking of ACE2, we treated the cells with bafilomycin A1 (Baf A1), which is an inhibitor of vacuolar H$^+$-ATPase to prevent reacidification of synaptic vesicles following exocytosis and block fusion of autophagosomes and lysosomes[27]. Our findings indicated that SARS-CoV-2 infection-induced ACE2 degradation could be rescued by Baf A1 treatment (Supplementary Fig. 4a), and ACE2 colocalized with early endosome at early stage of infection, but colocalized with late endosome at late stage of SARS-CoV-2 infection (Supplementary Fig. 4b, c).

**SUMOylation suppresses the autophagic degradation of ACE2**
To identify which degradation system dominantly contributes to the destabilization of ACE2, we observed that the PIAS4-mediated ACE2 upregulation could be blocked by 3-methyladenine (3-MA) (an autophagy inhibitor) or Baf A1, but not MG132 (a proteasome inhibitor) treatment (Fig. 4a and Supplementary Fig. 5a), suggesting that PIAS4 shut down the autophagic degradation of ACE2. Rapamycin

targets FKBP1A/FKBP12 to suppress mTORC1 activity, thus inducing autophagy, while Earle's balanced salts solution (EBSS), a saline solution buffer with physiological pH, induces starvation for autophagy activation[27]. *PIAS4* depletion potentiated the autophagic degradation of ACE2 induced by rapamycin (Fig. 4b and Supplementary Fig. 5b). We also observed that ACE2 underwent degradation under rapamycin treatment and EBSS-cultured condition in hNEPCs (Fig. 4c). Additionally, the autophagic degradation of ACE2 was abolished by Baf A1 treatment (Supplementary Fig. 5c). The PIAS4-triggered ACE2 stabilization was disappeared in *RB1CC1*- and *ATG13*-deficient cells, as well as *BECN1*- and *ATG5*-knockout (KO) cells, in which the canonical autophagy is significantly impaired (Fig. 4d, e and Supplementary Fig. 5d, e). We next assessed whether PIAS4 promotes the SUMOylation of ACE2 at K187 to prevent its autophagic degradation. Our data showed that the turnover rates of ACE2 were largely attenuated by K187R mutation upon autophagy activated conditions (Supplementary Fig. 5f–i). Collectively, these results suggested that PIAS4-mediated SUMOylation attenuates the degradation of ACE2 through autophagy.

To study how the cell surface receptor undergoes autophagic degradation, we further analyzed the macroautophagy with the treatment of SUMOylation inhibitor, observing SUMOylation could not significantly affect the LC3 lipidation, p62 degradation, Beclin-1 level and ATG5-12 conjugations (Supplementary Fig. 5j). These results indicated that the SUMOylation-mediated ACE2 stabilization was not caused by affecting the global macroautophagy. We detected the role of endocytosis in SUMOylation-mediated ACE2 stabilization by using multiple endocytic inhibitors and observed that ML-792 and GA-induced ACE2 degradation could be abolished by 3-MA treatment, but not affected by chlorpromazine (CPZ) (an inhibitor of clathrin-mediated endocytosis[28,29]), methyl-β-cyclodextrin (MβCD) (an inhibitor of caveolae-mediated endocytosis[30]) or cytochalasin D (an inhibitor of micropinocytosis[31,32]) (Fig. 4f). Moreover, the *PIAS4*-depletion-induced degradation of ACE2 was not influenced by endocytic inhibitors (Fig. 4g). We next found that SUMOylation inhibitor treatment suppressed the cytomembrane localization of ACE2 (Fig. 4h). Through separating the cytoplasmic and cell membranous components, we observed that SUMOylation promoted the cytomembrane distribution of ACE2 (Fig. 4i). The cytomembrane localization of ACE2 was decreased in *PIAS4*-deficet cells while it was enhanced in *SENP3*-depleted cells (Supplementary Fig. 5k). The ACE2 K187R mutant displayed the cytoplasmic distribution as the ACE2 ΔTM + CT mutant did (Supplementary Fig. 5l). The C-terminal of ACE2 named as collectrin-like-domain (CLD, aa 615–805), which contains a transmembrane (TM) helix (aa 740–762) and an intracellular tail (CT, aa 763–805)[33,34]. As the ACE2 ΔTM + CT mutant lacking the transmembrane domain, it could not be localized in the cell membrane.

Although numerous studies revealed that SUMOylation influences the subcellular localization of substrates[18,35–37], the underpinning

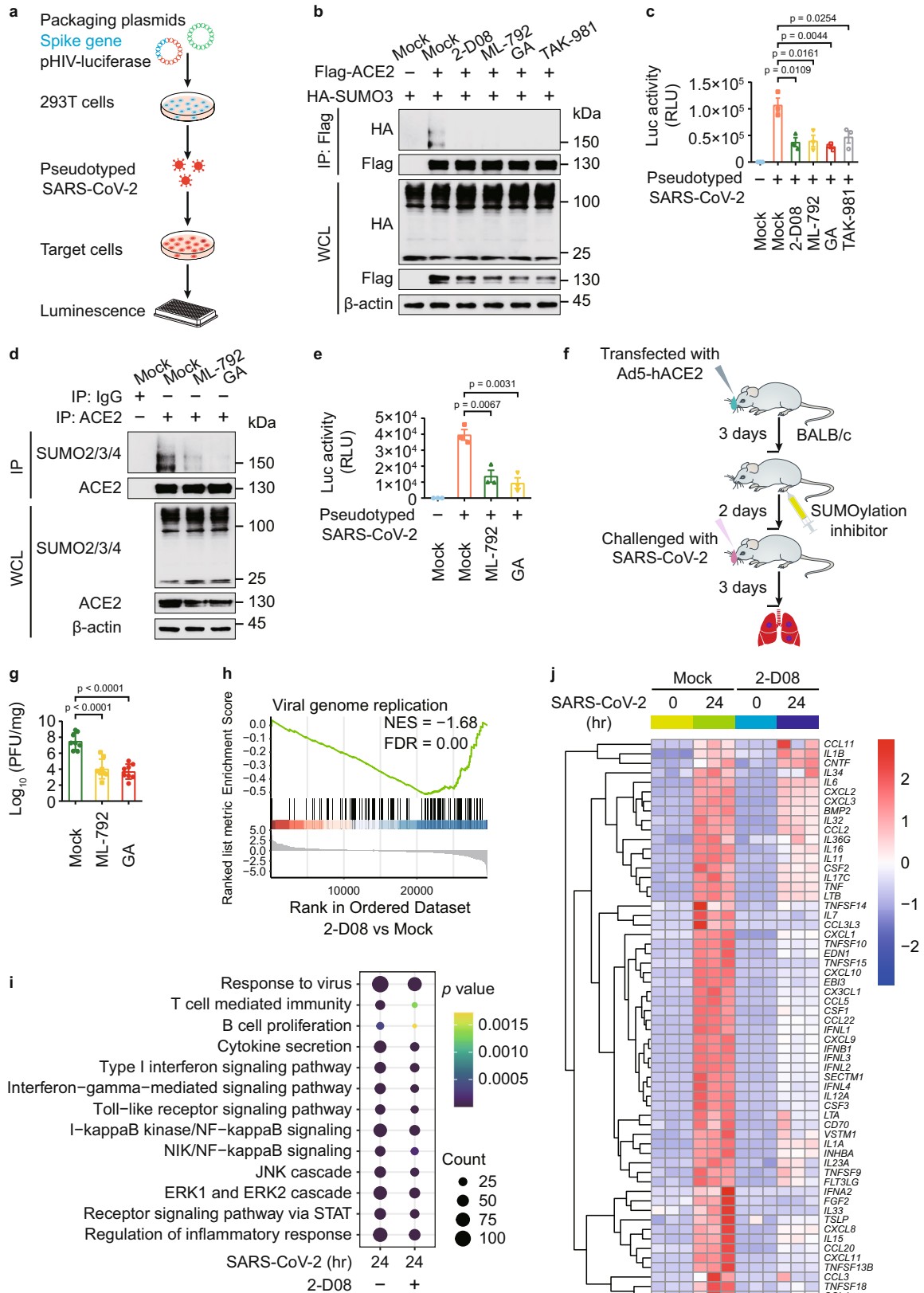

mechanism remains largely unclear. SLC6A19, also known as $B^0AT1$, is an amino acid transporter which localized in plasma membrane[38–40]. The presence of SLC6A19 hampers TMPRSS2 to get close to the cutting position on ACE2[41]. Additionally, ACE2 interacts with SLC6A19 directly for the cell surface expression[39]. To study how SUMOylation affects the subcellular distribution of ACE2, we checked the association between

ACE2 and SLC6A19 and found that SUMOylation promotes the ACE2-SLC6A19 interaction (Supplementary Fig. 5m). Together, these findings demonstrated that SUMOylated ACE2 is mainly localized in the cytomembrane, whereas deSUMOylated ACE2 tends to be distributed in the cytoplasm, where it can be degraded through selective autophagy.

**Fig. 2 | Manipulation ACE2 SUMOylation contributes to the SARS-CoV-2 infection. a** The strategy to acquire SARS-CoV-2 pseudovirus based on the HIV-1 three-plasmid packaging system. **b** Immunoblotting of 293 T cells transfected with plasmids of Flag-ACE2 and HA-SUMO3 in the presence of 2-D08 (200 μM), ML-792 (10 μM), GA (5 μM), or TAK-981 (5 μM). **c** Luciferase activity of Calu-3 cells treated with SARS-CoV-2 S pseudotyped virus for 48 hr, along with SUMOylation inhibitor treatment. **d** Immunoprecipitation and immunoblot analysis of human alveolar epithelial cells (AECs) treated with ML-792 (10 μM) or GA (5 μM). **e** Luciferase activity of SARS-CoV-2 S pseudotyped virus-infected AECs, along with ML-792 (10 μM) or GA (5 μM) treatment. **f** Schematic illustration of the adenoviral transduction-based mouse model to study SARS-CoV-2 infection. **g** Ad5-hACE2-transduced BALB/c mice were intranasally infected with SARS-CoV-2 ($1 \times 10^5$ PFU) in 50 mL of DMEM for 3 days. The lungs of mice were harvested for viral plaque assay. Titers are expressed as PFU/g lung tissue. **h** GSEA analysis of differentially expressed genes (DEGs) in Calu-3 cells infected with SARS-CoV-2 (MOI = 0.5) for 24 h with or without 2-D08 (100 μM) treatment. FDR (q-value) was shown. **i** Dot-plot visualization of enriched GO terms showing the enriched genes upregulated by SARS-CoV-2 with or without 2-D08 treatment comparing with negative control. The ordinate is the GO term description. Bubble size represents the number of DEGs in GO classification; the enrichment *P*-value is calculated by Fisher exact test; and different color represents different *P*-value. **j** Heatmap showing the transcription levels of DEGs belonging to GO annotations for cytokine activity and chemokine activity (GO: 0005125 and GO: 0008009) upregulated by SARS-CoV-2. In **c** and **e**, all error bars, mean values ± SEM, *P*-values are determined by unpaired two-tailed Student's *t*-test of *n* = 3 independent biological experiments. In **g**, all error bars, mean values ± SD, *P*-values are determined by unpaired two-tailed Student's *t*-test (*n* = 8 independent biological mice per group). For **b**, **d**, similar results are obtained by three independent biological experiments. Source data are provided as a Source data file.

## TOLLIP targets ACE2 for selective autophagic degradation

Since cargo receptors are essential to deliver substrates for selective autophagic degradation[14,16]. We found that ACE2 interacted with TOLLIP, but not other cargo receptors (Fig. 5a), and further detected the endogenous association between ACE2 and TOLLIP (Supplementary Fig. 6a). Immunofluorescence analysis of the lung from transgenic hACE2 mice indicated that TOLLIP-ACE2 interaction was largely elevated upon SARS-CoV-2 infection (Fig. 5b, c). Co-immunoprecipitation experiments revealed that SARS-CoV-2 infection enhanced the ACE2-TOLLIP association. Intriguingly, the interaction between ACE2 and PIAS4 was decreased while the association between ACE2 and SENP3 was increased (Fig. 5d). By knocking down the expression of *TOLLIP*, the autophagic degradation of ACE2 was abolished (Supplementary Fig. 6b). PIAS4 disrupted the association between ACE2 and TOLLIP, whereas SENP3 facilitated the interaction between ACE2 and TOLLIP (Supplementary Fig. 6c, d). Consistently, PIAS4- and SENP3-mediated ACE2 regulation was abrogated in *TOLLIP*-deficient cells (Supplementary Fig. 6e–h). Altogether, these results indicated that PIAS4 and SENP3 modulate the recognition of ACE2 by TOLLIP to affect the selective autophagic degradation of ACE2.

## TOLLIP-directed selective autophagy protects against SARS-CoV-2

We further found that *TOLLIP* depletion increased the levels of SARS-CoV-2 S pseudotyped virus (Fig. 5e). Moreover, *TOLLIP* deficiency resulted in the accumulation of ACE2 and the increased SARS-CoV-2 load (Fig. 5f, g). We next performed RNA-seq analysis to examine the role of TOLLIP-directed ACE2 degradation under SARS-CoV-2-infected conditions. DEGs were determined by comparing the transcriptomes of SARS-CoV-2-infected cells with *TOLLIP* depletion with that of scramble siRNA-treated infection group as control. GSEA indicated upregulation of genes was associated with 'viral genome replication' in *TOLLIP*-deficient cells (Fig. 5h). The gene-ontology enrichment analysis showed that *TOLLIP* depletion upregulated a variety of pathways under SARS-CoV-2 infection (Supplementary Fig. 6i). The heatmap of genes associated with cytokines and chemokines activity revealed that *TOLLIP* deficiency potentiated SARS-CoV-2 infection (Fig. 5i). The proliferation of SARS-CoV-2 and host antiviral immune responses were increased in *TOLLIP*-deficient cells (Supplementary Fig. 6j). All these results suggested that TOLLIP-dependent autophagic degradation of ACE2 protects against SARS-CoV-2.

## TOLLIP recognizes ACE2 in a K48-linked ubiquitination-dependent manner

Since SUMOylation prompted the stabilization of ACE2, we next examined if SUMOylation inhibits the recognition of ACE2 by cargo receptor TOLLIP, finding that ACE2-TOLLIP association was increased upon 2-D08 treatment (Fig. 6a and Supplementary Fig. 7a). TOLLIP consists of a Tom1-binding domain (TBD) (aa 1–54), a phospholipid-binding C2-domain (aa 55–181) harboring two LC3-interacting regions (LIRs), and a ubiquitin-binding CUE domain (aa 219–274) (Fig. 6b). To further elucidate the mechanism of ACE2 recognition, we generated TOLLIP R78A or M240A/F241A (CUE domain mutant 1), L267A/L268A (CUE domain mutant 2) constructs, which prevent the binding of TOLLIP with phosphatidylinositol 3-phosphate (PI3P) and PI(4,5)P₂ or ubiquitin, respectively[42,43]. Since TOLLIP CUE mutants cannot interact with ACE2 (Fig. 6c), we wondered that ubiquitination may function as a recognition signal for TOLLIP-dependent autophagic degradation of ACE2. We performed MS analysis to investigate the ubiquitination of purified ACE2, and identified a specific K48-linked polyubiquitin peptide (Fig. 6d). The findings of MS data were also validated by co-immunoprecipitation assay and the K48-linked ubiquitination of ACE2 was remarkably elevated upon 2-D08 treatment (Fig. 6e). We next employed the plasmid expressing HA-K48R-Ub, which features a K48 to R48 mutation, and observed that the ubiquitination of ACE2 with other linkage type was not influenced after 2-D08 treatment (Supplementary Fig. 7b). Collectively, these results indicated that K48-linked ubiquitin chains on ACE2 provide recognition signal for TOLLIP.

To study whether SUMOylation dampens the K48-linked ubiquitination of ACE2, we first studied the role of PIAS4 in ACE2 ubiquitination and found that the K48-linked ubiquitination of ACE2 was increased by *PIAS4* silencing (Fig. 6f). Additionally, the K48-linked ubiquitination of ACE2 K187R mutant was barely observed (Fig. 6g). The SUMOylation of ACE2 was deceased, while the K48-linked ubiquitination of ACE2 was enhanced in EBSS-cultured autophagic condition (Supplementary Fig. 7c). Immunofluorescence analysis revealed that the colocalization of K48-linked ubiquitin chains and ACE2 was enhanced in *SENP3*-deficient cells (Supplementary Fig. 7d, e). ACE2 ubiquitinated at K187 was further confirmed by MS analysis (Fig. 6h). The interaction between TOLLIP and ACE2 K187R mutant was impaired (Fig. 6i), and the ACE2 K187R mutant had slower turnover rates (Fig. 6j, k). These findings suggested that K187 is an essential site for both SUMO3 and K48-linked polyubiquitin modification of ACE2. Once the SUMO3 conjugation of ACE2 at K187 is decreased, the K48-linked ubiquitination at the same lysine functions as a critical K48-linked ubiquitination site, thus mediating the destabilization of ACE2 through TOLLIP recognition.

We also studied whether SUMOylation regulates the association between ACE2 and SARS-2-S, observing that the ACE2-Spike association remained unchanged with the existence of SUMOylation inhibitor (Supplementary Fig. 7f). Moreover, we found that the ACE2 K187R mutant, in which SUMOylation and ubiquitination were abrogated, could still interact with the spike protein (Supplementary Fig. 7g). Consist with the result that 2-D08 prompts the degradation of WT ACE2, but not K187R ACE2 mutant, the suppression of SARS-CoV-2 infection induced by 2-D08 treatment was abolished in K187R ACE2-expressing cells (Supplementary Fig. 7h). Altogether, our results

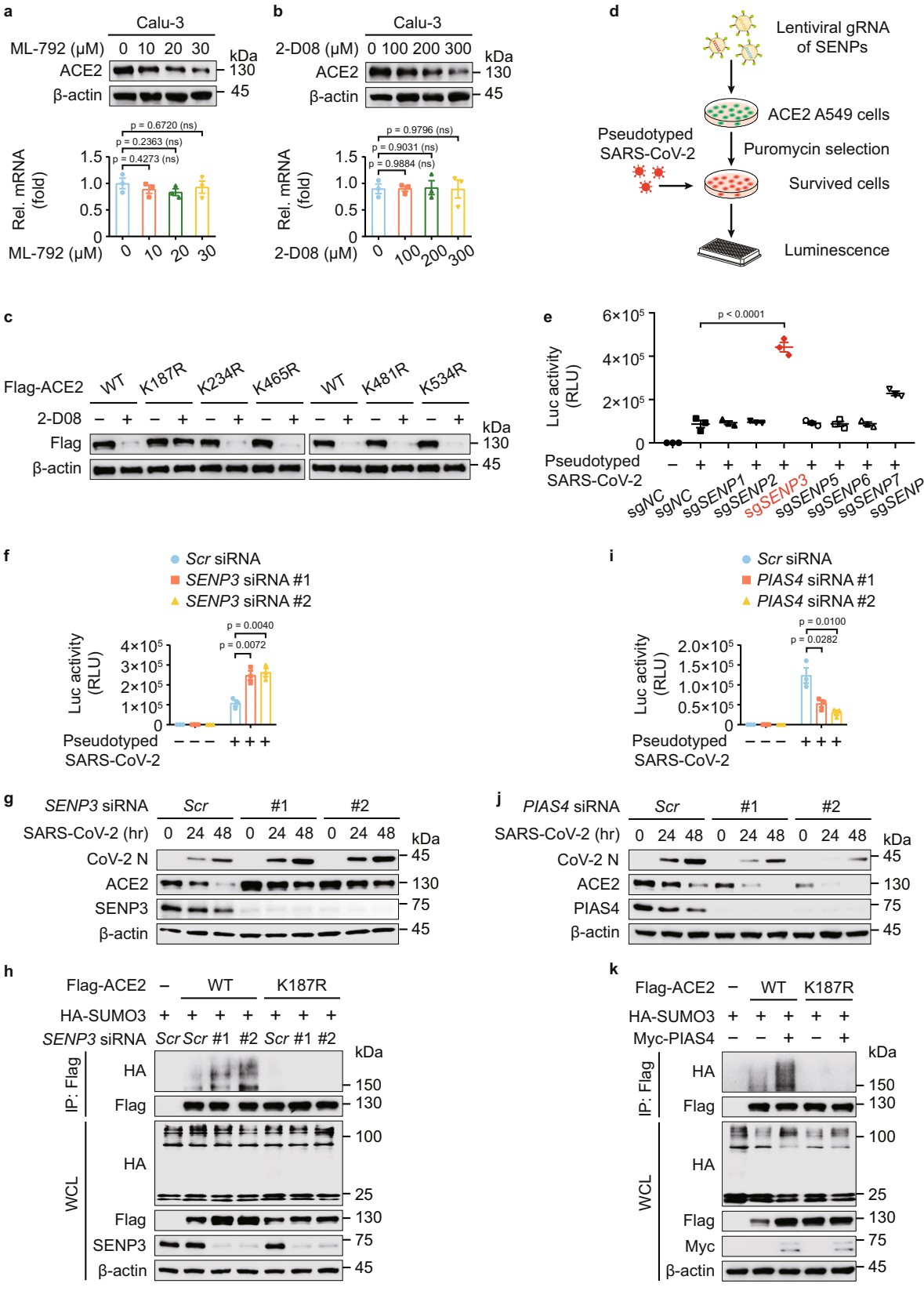

suggested that SUMOylation facilitates SARS-CoV-2 infection by promoting the protein stability of ACE2.

## Discussion

The recently emerged human pandemic coronavirus SARS-CoV-2, the causative agent of COVID-19, has caused substantial morbidity and mortality worldwide[44,45]. To defeat SARS-CoV-2, an urgent demand of unmasking the host–virus biology would give insights into antiviral treatment. The viral envelope of SARS-CoV-2 coated by spike protein trimers targets ACE2 for viral entry[46]. ACE2 is a critical carboxypeptidase in the regulation of blood pressure, the balance of salt and water, the uptake of amino acid in small intestine, as well as the glucose

**Fig. 3 | SUMOylation promotes the stabilization of ACE2. a**, **b** Protein abundance (up) and mRNA level (down) of ACE2 in Calu-3 cells treated with indicated concentration of ML-792 (**a**) or 2-D08 (**b**) for 24 h. **c** 293 T cells expressing wild-type (WT) Flag-ACE2 or its indicated mutants were treated with 2-D08 (200 μM) for 24 h. Different amounts of vectors for WT and mutants of Flag-ACE2 were transfected into 293 T cells for equal expression. The lysates were analyzed with immunoblotting. **d** Overview of the screening for functional deSUMOylating enzymes in human ACE2-expressing A549 cells with the infection of SARS-CoV-2 S pseudotyped virus. **e** Luciferase activity in control and deSUMOylating enzyme depletion hACE2-expressing A549 cells challenged with SARS-CoV-2 S pseudotyped virus. **f** Calu-3 cells transfected with scramble or *SENP3*-specific siRNAs were infected with SARS-CoV-2 S pseudotyped virus for 48 h, the protein extracts were harvested for luciferase assay. **g** Calu-3 cells were transfected with scramble or *SENP3*-specific siRNAs treated with SARS-CoV-2 (MOI = 0.5) for indicated time points. The protein extracts

were harvested for immunoblot analysis. **h** 293 T cells transfected with scramble or *SENP3*-specific siRNAs were then transfected with plasmids encoding Flag-ACE2 and HA-SUMO3. The protein extracts were harvested for immunoprecipitation and immunoblot analysis. **i** Luciferase activity of Calu-3 cells transfected with scramble or *PIAS4*-specific siRNAs, along with SARS-CoV-2 S pseudotyped virus infection for 48 h. **j** Lysates of Calu-3 cells transfected with scramble or *PIAS4*-specific siRNAs treated with SARS-CoV-2 (MOI = 0.5) for indicated time points. The protein extracts were harvested for immunoblot analysis. **k** Lysates of 293 T cells transfected with vectors of HA-SUMO3 and Flag-ACE2 or its K187R mutant along with vector expressing Myc-PIAS4 were immunoprecipitated with anti-Flag and immunoblotted with anti-HA. In **a**, **b**, **e**, **f**, **i**, all error bars, mean values ± SEM, *P*-values are determined by unpaired two-tailed Student's *t*-test of *n* = 3 independent biological experiments. For **a**–**c**, **g**, **h**, **j**, **k**, similar results are obtained by three independent biological experiments. Source data are provided as a Source data file.

homeostasis and the function of pancreatic β-cell[47]. Herein, we investigated the protein interaction and regulation of ACE2 through mass spectrometry analysis, and verified that SUMO pathway is involved in the system-wide identification. We elucidated that ACE2 receptor is a target for SUMO modification, and specific covalent attachment of SUMO3 to ACE2 mediates the ACE2 stabilization. PIAS4 catalyzes the conjugation of SUMO3 to ACE2 at K187, while SENP3 removes the attachment of SUMO3 to ACE2. The association between ACE2 and PIAS4/SENP3 could be detected under basal levels in the soluble cytosol (Supplementary Fig. 3j). Thus, the SUMOylation system controls the protein level of ACE2 at the basal conditions to further influence the subsequent SARS-CoV-2 infection. Since the interaction between ACE2 and SENP3 was enhanced while the interaction between ACE2 and PIAS4 was decreased under SARS-CoV-2 infection (Fig. 5d), we hypothesized that there may be a competitive interaction of PIAS4 and SENP3 with ACE2 and the post-translational modification or the upstream activation signaling lead to the altered association between SENP3 and ACE2 under SARS-CoV-2 infection. As SARS-CoV-2 infection causes the dysregulation of mTOR signaling and mitosis[48,49], which are essential for the modification and function of SENP3[50–52], perhaps the phosphorylation of SENP3 could be further influenced, thereby affecting the association between ACE2 and PIAS4/SENP3 during SARS-CoV-2 infection. Since SUMO modification works as a positive mediator in the tightly control of ACE2 stability, reversible SUMOylation of ACE2 can be considered as pharmacological targets in antiviral therapy of COVID-19.

Notably, SUMOylation is a post-translational modification regulating numerous cellular processes[21]. During viral infection, SUMO-mediated regulation is involved in the virus replication and host type I interferon responses[53–55]. Our study indicated that treatment with global SUMOylation inhibitors resulted in the destabilization of ACE2 and the reduced SARS-CoV-2 infection in cells. Moreover, global SUMOylation inhibitors suppressed SARS-CoV-2 infection in Ad5-hACE2-transduced mice. TRIM38 has been reported to prompt the SUMOylation of RIG-I and MDA5 to activate antiviral immune responses, and TRIM38-directed SUMOylation also promotes the stability of cGAS and STING to regulate DNA virus-induced type I interferon signaling[53,56]. However, IRF3 and IRF7 SUMOylation lead to a reduction of type I interferon production[57]. STAT1 SUMOylation leads to a decrease of STAT1 phosphorylation and selectively reduces transcription of interferon-gamma target genes[54]. Since SUMO orthologs function both as positive and negative factors for viral replication and host innate immune responses, the antiviral effects of global SUMOylation inhibitors in mice might not solely be due to prevention of ACE2 SUMOylation. However, transduced human ACE2 is pivotal for SARS-CoV-2 infection in our mice model, thus ACE2 regulation by SUMO inhibitors plays an important role in this infection process. Screening of specific SUMOylation inhibitor can solve this problem and local drug delivery should be developed in the future clinal research.

Although the majority of SUMO targets are localized in the nucleus and are thought to undergo rapid cycles of SUMO conjugation and deconjugation[20], our knowledge towards SUMOylation of cytoplasmic proteins is quite limited. Our results indicated that SUMO3 modification impedes the K48-linked ubiquitination and autophagic degradation of ACE2. We suggested that at least three forms of ACE2 may exist in cells: ACE2 SUMOylated at K187, ACE2 modified by K48-linked poly-ubiquitination at K187, and unmodified ACE2. SUMOylation affects the cellular distribution of ACE2 and blocks the K48-linked ubiquitination of ACE2 at K187. As the suppression of SARS-CoV-2 infection induced by 2-D08 treatment was abolished in K187R ACE2-expressing cells (Supplementary Fig. 7h), indicating that SUMOylation and stabilization of ACE2 participate in the antiviral regulation of SARS-CoV-2 infection. Mounting evidence has supported that ubiquitination of substrates plays a crucial role in selective autophagy[15,58]. We observed the regulatory roles of PIAS4 and SENP3 are abrogated in *TOLLIP* depletion cells and revealed that cargo receptor TOLLIP arrests ACE2 to deliver it to autophagosome for subsequent degradation. The TOLLIP-ACE2 association relays on the CUE domain of TOLLIP, and TOLLIP delivers the K48-linked ubiquitylated ACE2 to autophagosomes for degradation.

TOLLIP has also been reported in facilitating endosomal organization and cargo trafficking, which is dependent on the interaction with ubiquitinated cargo receptor p62 and ubiquitin E3 ligase RNF26[43,59]. Our findings revealed that the SUMOylation inhibitor treatment-induced ACE2 degradation could not be rescued by endocytosis inhibitors (Fig. 4f, g), and SUMOylation-mediated ACE2 stabilization was not caused by affecting the global macroautophagy (Supplementary Fig. 5j), suggesting SUMOylation mainly maintains the stabilization of ACE2 through selective autophagy pathway. Recent research indicated that ORF3a of SARS-CoV-2 perturbs the assembly of the SNARE complex mediated by HOPS-complex to block autolysosome formation[60]. SARS-CoV-2 infection activates negative mediators while reduced positive mediators of autophagy, and inhibits autophagic flux by downregulation of Beclin-1/ATG14-dependent autophagosome-lysosome fusion[61]. Moreover, SARS-CoV-2 NSP6 targets hybrid pre-autophagosomal structure (HyPAS) apparatus to inhibit prophagophore formation[62]. SARS-CoV-2 particles colocalize with lipid droplets for their replication[63]. The interplay between autophagy, lipophagy, and SARS-CoV-2 infection is important for the antiviral therapy of COVID-19, and the detailed regulatory mechanisms of autophagy and trafficking in ACE2-mediated SARS-CoV-2 infection warrant further dissection.

Previous evidence has discovered that TOLLIP plays an essential role in the interleukin-1 receptor and Toll-like receptor-mediated innate immune signaling pathways[64]. As a modulator of the immune pathway, TOLLIP indirectly controls the amounts of antimicrobial peptides to prevent infection[65,66]. Genetic variation in *TOLLIP* gene is concerned to link with the susceptibility to intracellular infections[67]. We found that the SARS-CoV-2 infection and the consequent host

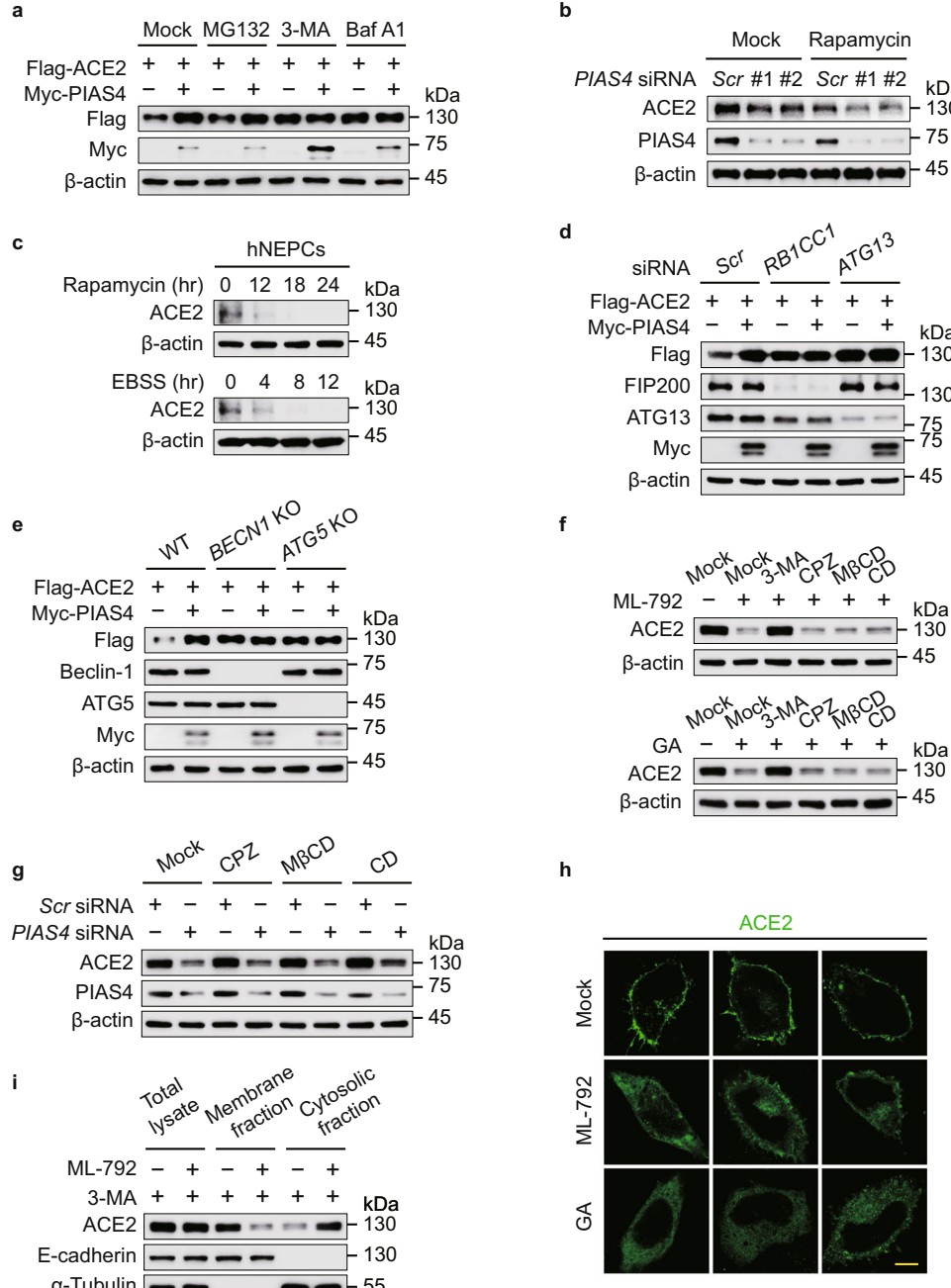

**Fig. 4 | SUMO3 conjugation attenuates ACE2 degradation through autophagy.**
**a** 293 T cells were transfected with plasmid encoding Flag-ACE2 together with Myc-PIAS4 vector treated with MG132 (10 μM), 3-methyladenine (10 mM), or bafilomycin A1 (0.2 μM) for 6 h. The cell extracts were analyzed by immunoblotting. **b** Lysates of Calu-3 cells transfected with scramble or *PIAS4*-specific siRNA and treated with rapamycin (250 nM) for 24 h, were collected for immunoblot analysis. **c** hNEPCs were treated with rapamycin (250 nM) (up) or EBSS (down) with indicated time points, and the proteins were harvested for immunoblot analysis. **d** 293 T cells transfected with scramble siRNA along with *RB1CC1-* or *ATG13*-specific siRNAs were transfected with plasmids encoding Flag-ACE2 and Myc-PIAS4, the lysates were analyzed by immunoblotting. **e** Wild-type (WT), *BECN1-* and *ATG5*-knockout (KO) 293 T cells were transfected with vectors of Flag-ACE2 and Myc-PIAS4, the lysates were subjected to immunoblot analysis. **f** Calu-3 cells were treated with ML-792 (10 μM) (up) or ginkgolic acid

(GA) (5 μM) (down) together with 3-MA (10 mM), chlorpromazine (CPZ) (10 μM), methyl-β-cyclodextrin (MβCD) (1 mM), or cytochalasin D (CD) (2 μM), and the proteins were harvested for immunoblot analysis. **g** Immunoblot analysis of Calu-3 cells transfected with scramble or PIAS4-specific siRNAs and treated with chlorpromazine (CPZ) (10 μM), methyl-β-cyclodextrin (MβCD) (1 mM), or cytochalasin D (CD) (2 μM). **h** HeLa cells transfected with Flag-ACE2 along with ML-792 (10 μM) or ginkgolic acid (GA) (5 μM) treatment, followed by labeling of ACE2 (green) with specific antibodies. Scale bar, 20 μm. Representative confocal microscopy images from *n* = 3 independent biological samples for each group. **i** Total, cytosolic, and membrane fractions of Calu-3 cells pretreated with DMSO or ML-792 (10 μM) were immunoblotted with antibodies directed against ACE2, a cytosol marker (α-tubulin), or a PM marker (E-cadherin). In **a**–**g** and **i**, similar results are obtained by three independent biological experiments. Source data are provided as a Source data file.

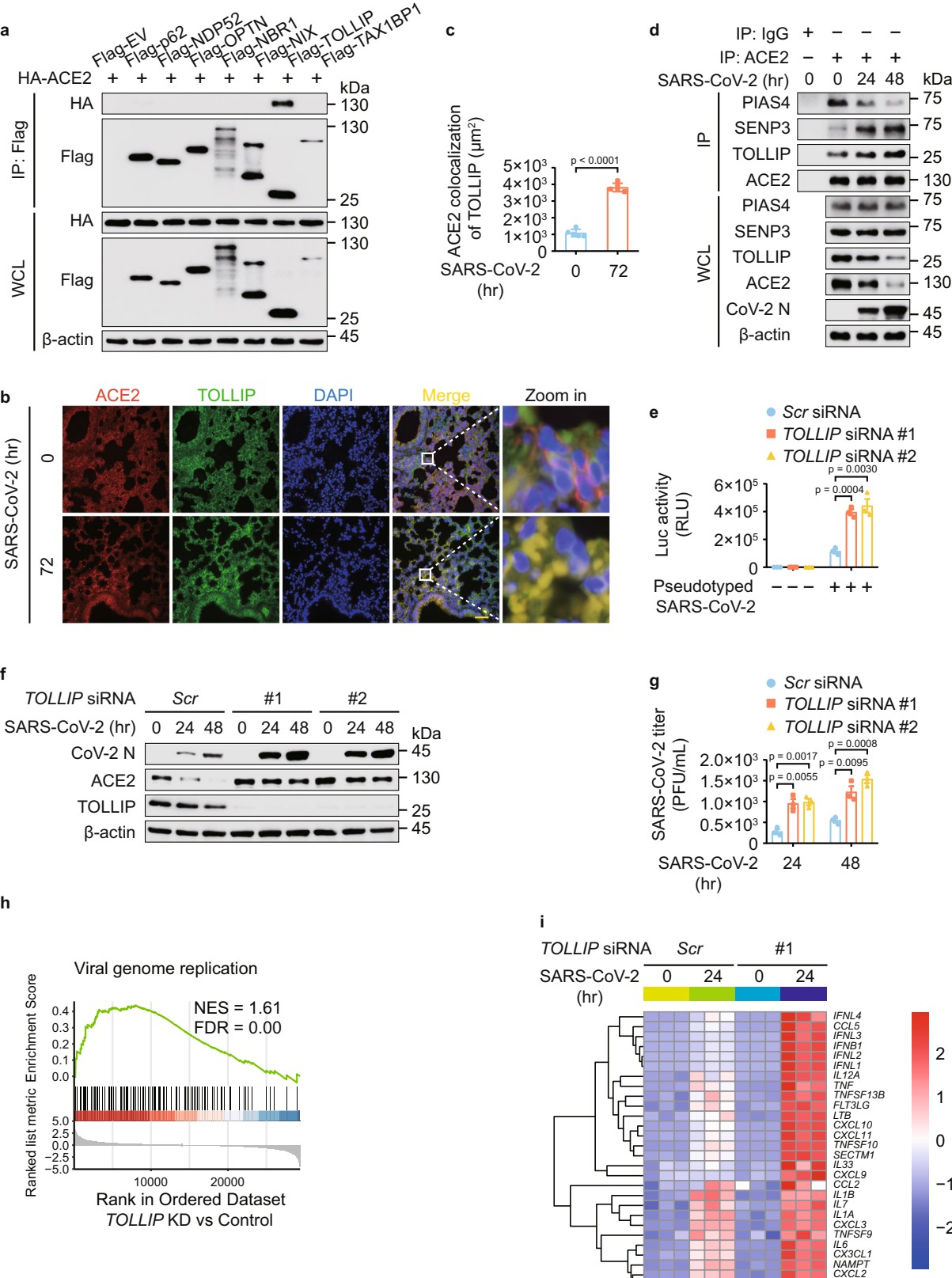

antiviral immune responses were largely increased in *TOLLIP*-deficient cells. Our data proved that ubiquitinated ACE2 serves as a new substrate for TOLLIP-mediated selective autophagic degradation to influence the infection of SARS-CoV-2 in a ubiquitin signal dependent way. The regulatory roles of macroautophagy, secretory autophagy, as well as SUMOylation of host immune factors and viral proteins in

contributing to SARS-CoV-2-triggered immune responses merit further study in the future. Altogether, we demonstrated that dynamic SUMOylation strictly manipulates the stabilization of ACE2 protein and contributes to the infection of SARS-CoV-2. Our findings uncovered a regulatory mechanism underpinning TOLLIP-mediated selective autophagic degradation of ACE2 based on dynamic changes between

**Fig. 5 | TOLLIP is involved in ACE2 degradation and the associated SARS-CoV-2 infection. a** Immunoprecipitation and immunoblot analysis of 293 T cells transfected with vectors of HA-ACE2 and indicated Flag-tagged cargo receptors. **b** Representative ACE2 (red) and TOLLIP (green) immunofluorescent staining in lung from transgenic hACE2 mice upon SARS-CoV-2 ($1 \times 10^5$ TCID50/mL) infection for 72 h from $n = 5$ independent biological mice per group. Scale bar, 50 μm. **c** Area of ACE2 and TOLLIP colocalization. **d** Calu-3 cells were challenged with SARS-CoV-2 (MOI = 0.5) for indicated time points and the protein extracts were immunoprecipitated and analyzed by immunoblotting. **e** Calu-3 cells transfected with scramble or *TOLLIP*-specific siRNAs were infected with SARS-CoV-2 S pseudotyped virus for 48 h, the protein extracts were harvested for luciferase assay. **f** Immunoblot analysis of Calu-3 cells transfected with scramble or *TOLLIP*-specific siRNAs and infected with SARS-CoV-2 (MOI = 0.5) for indicated time points. **g** Plaque titration of

SARS-CoV-2 in supernatants of scrambled or *TOLLIP*-specific siRNAs transfected Calu-3 cells infected with SARS-CoV-2 (MOI = 0.5). **h** GSEA analysis of genes expressed differentially in Calu-3 cells with or without *TOLLIP*-specific siRNA transfection upon SARS-CoV-2 (MOI = 0.5) infection for 24 h. FDR (q-value) was shown. **i** Heatmap indicating the transcription levels of DEGs induced by SARS-CoV-2 (MOI = 0.5) infection for 24 h with different treatments, and the genes belonging to GO annotations for cytokine activity and chemokine activity (GO: 0005125 and GO: 0008009). In **c**, all error bars, mean values ± SD of five images from $n = 5$ independent biological mice for each group. *P*-values are indicated by two-tailed unpaired Student's *t*-test. In **e**, **g**, all error bars, mean values ± SEM, *P*-values are determined by unpaired two-tailed Student's *t*-test of $n = 3$ independent biological experiments. For **a**, **b**, **d**, **f**, similar results are obtained by three independent biological experiments. Source data are provided as a Source data file.

SUMOylation and ubiquitination, which affects the entry of SARS-CoV-2 and points out potential therapy targets for SARS-CoV-2.

## Methods

### Animals

C57BL/6 transgenic hACE2 mice and wild-type BALB/c mice, both male and female, between the age of 8–12 weeks, were acquired from Guangzhou Medical Laboratory Animal Center of China. Mice were housed in a specific pathogen-free facility with a temperature of 20–26 °C, humidity between 40 and 70%, under 12 h light/dark cycle at Sun Yat-sen University. All animal experimental protocols were approved by the Institutional Animal Care and Use Committee (IACUC) of Sun Yat-sen University [Authorization number: SYXK (YUE) 2017-0175, Guangzhou, China].

### Cells and culture conditions

HEK293T (#GNHu17), HeLa (#TCHu187), A549 (#TCHu150), HepG2 (#TCHu72), Calu-3 (#TCHu157), and Vero E6 (#GNO17) cells were obtained from National Collection of Authenticated Cell Cultures (Shanghai, China). HEK293T, HeLa, A549, HepG2, and Vero E6 cells were cultured in DMEM medium (Corning, cat. 10-013-CV) with 10% fetal bovine serum (FBS) [Gibco, cat. 10099141] and 1% glutamine (Gibco, cat. 35050061). Calu-3 cells were cultured in MEM (Gibco, cat. 41090036) supplemented with 1% NEAA (Gibco, cat. 11140050), 1% sodium pyruvate (Gibco, cat. 11360070), and 10% FBS. hUVECs (#GDC0635) purchased from China Center for Type Culture Collection (Wuhan, China), were maintained in Endothelial Cell Medium (ScienCell, cat. 1001) supplemented with 5% FBS and endothelial cell growth supplement (Corning, cat. 356006). Human alveolar epithelial cells (AECs) (#3200) obtained from ScienCell Research Laboratories (San Diego, USA), were cultured in Alveolar Epithelial Cell Medium (ScienCell, cat. 3201). Primary nasal epithelial cells (hNEPCs) were obtained from patients with nasal polyps, who underwent functional endoscopic sinus surgery. Approval for the sample collection and cell culture experiment was obtained from the Institutional Review Board of The First Affiliated Hospital, Sun Yat-sen University, with ethics approval number [2017]303. All patients provided written informed consent, in accordance with the Declaration of Helsinki and as recommended by the First Affiliated Hospital, Sun Yat-sen University. hNEPCs were cultured using a modified method based on the previous described[68]. To active starvation-induced autophagy, cells were washed with phosphate-buffered saline (PBS) for at least two times and incubated in EBSS (Gibco, cat. 14155063). All cells were cultured at 37 °C with 5% $CO_2$.

### Transduction and infection of mice

BALB/c mice were transduced intranasally with $2.5 \times 10^8$ FFU of Ad5-hACE2 or Ad5-Empty in 75 μL DMEM after lightly anesthetized with isoflurane. Five days later, the mice were infected intranasally with $1 \times 10^5$ PFU of SARS-CoV-2 in 50 μL DMEM. All experiments with SARS-

CoV-2 infection were performed in the Biosafety Level 3 Laboratory of Guangzhou Customs District Technology Center.

### Antibodies and reagents

Horseradish peroxidase (HRP) anti-Flag (M2; #A8592), and anti-β-actin (#A1978) were acquired from Sigma-Aldrich. HRP anti-hemagglutinin (HA; #12013819001) and anti-c-myc (Myc; #11814150001) were obtained from Roche Applied Science. Anti-ACE2 (#21115-1-AP), anti-PIAS4 (#14242-1-AP), anti-FIP200/RB1CC1 (#17250-1-AP), anti-Beclin-1 (#11306-1-AP), anti-ATG5 (#10181-2-AP), anti-p62/SQSTM1 (#18420-1-AP), anti-TOLLIP (#11315-1-AP), anti-E-cadherin (#20874-1-AP), anti-α-Tubulin (#11224-1-AP), anti-Syntaxin 6 (#60059-1-Ig), anti-mouse (#SA00001-1), and anti-rabbit (#SA00001-2) were acquired from Proteintech. K48-linkage specific polyubiquitin antibody (#4289), anti-ULK1 (D8H5) [#8054], anti-ATG13 (E1Y9V) [#13468], anti-ACE2 (#4355), anti-EEA1 (C45B10) [#3288], and anti-Rab7 (D95F2) [#9367] were purchased from Cell Signaling Technology. Anti-SENP3 (#sc-133149), anti-SUMO1 (#sc-5308), anti-SUMO2/3/4 (#sc-393144), and anti-SEC61β (#sc-393633) were from Santa Cruz Biotechnology. Anti-CoV-2-N (#40588-T62) was acquired from Sino Biological. Goat anti-rabbit IgG (H + L) highly cross-adsorbed secondary antibody, Alexa Fluor 488 (#A-11034); goat anti-mouse IgG (H + L) highly cross-adsorbed secondary antibody, Alexa Fluor 488 (#A-11029); goat anti-rabbit IgG (H + L) cross-adsorbed secondary antibody, Alexa Fluor 568 (#A-11011); goat anti-mouse IgG (H + L) highly cross-adsorbed secondary antibody, Alexa Fluor 568 (#A-11004); goat anti-mouse IgG (H + L) cross-adsorbed secondary antibody, Alexa Fluor 594 (#A-11005); and goat anti-mouse IgG (H + L) cross-adsorbed secondary antibody, Alexa Fluor 633 (#A-21050) were purchased from Invitrogen. All primary antibodies were used at a dilution of 1:1000 for immunoblotting, 1:400 for immunoprecipitating and 1:200 for immunofluorescent staining. All secondary antibodies were used at a dilution of 1:5000 for immunoblotting and 1:400 for immunofluorescent staining. ML-792 (#HY-108702), 2-D08 (#HY-114166) and ginkgolic acid (GA) (#HY-N0077) were from MedChemExpress (MCE). MG132 (#C-2211), 3-methyladenine (#M9281), bafilomycin A1 (#H2714), N-Ethylmaleimide (NEM) (#E3876), CPZ (#C0982), MβCD (#C4555), and cytochalasin D (CD) (#C2618) were purchased from Sigma. Subasumstat (TAK-981) [#S8829] was acquired from Selleck.

### Plasmids and siRNA transfection

The pcDNA3.1-HA-SUMO1-4, SENP3, TOLLIP; pcDNA3.1-Flag-p62, NDP52, OPTN, NBR1, NIX, TOLLIP, TAX1BP1; pcDNA3.1-ACE2-HA, and pcDNA3.1-ACE2-Flag as well as its indicated truncation mutants were constructed by standard molecular biology technique. The pRK5-HA-Ubiquitin-K48 (#17605) and pRK5-HA-Ubiquitin-K48R (#17604) were purchased for Addgene. Expression plasmids for PIAS1, PIAS2α, PIAS2β, PIAS3, and PIAS4 were provided by Dr. Hong Tang (University of Chinese Academy of Sciences). Construct coding for ACE2 was

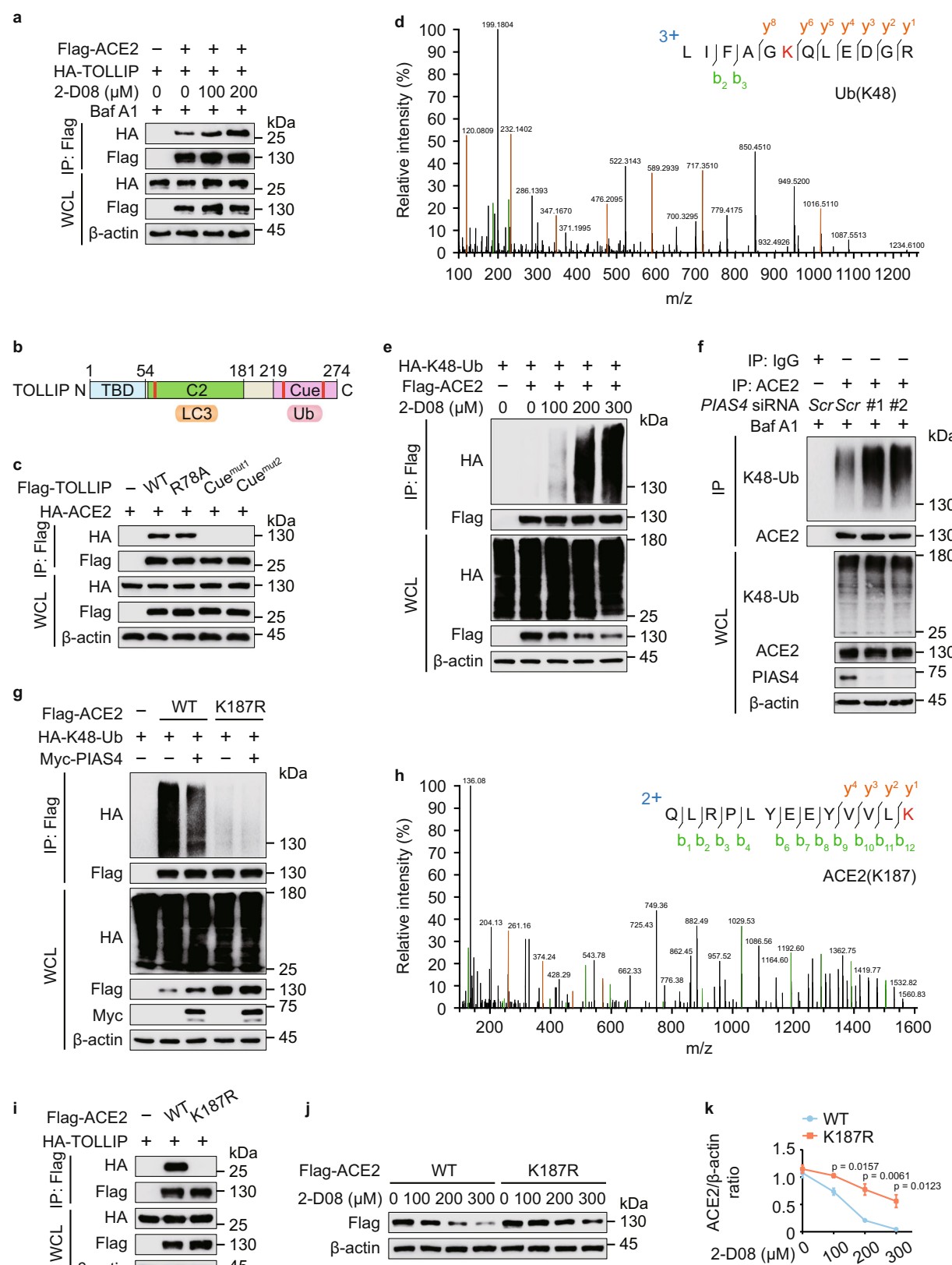

cloned into the FG-EH-DEST (provided by Dr. Xiaofeng Qin's laboratory) for retroviral expression. Plasmids encoding K187R, K234R, K465R, K481R, and K534R mutants of ACE2 were generated using pcDNA3.1-ACE2-Flag as a template by site-directed mutagenesis using the Mut Express® II Fast Mutagenesis Kit V2 (Vazyme, cat. C214-01), and the primers could be found in Supplementary Table 2. Transient

transfection into HEK293T cells was conducted using Lipofectamine 2000 (Invitrogen, cat. 11668019). Chemically synthesized 21-nucleotide siRNA duplexes were acquired from Sangon Biotech and transfected using Lipofectamine RNAiMAX (Invitrogen, cat. 13778030). Detailed information of all siRNA sequences could be found in Supplementary Table 3.

**Fig. 6 | SUMOylation dampens the K48-linked ubiquitination of ACE2.**
**a** 293 T cells were transfected with plasmids encoding Flag-ACE2 and HA-TOLLIP, and treated with indicated concentrations of 2-D08 for 24 h. The cells cultured with Baf A1 (0.2 µM) for 6 h were immunoprecipitated with α-FLAG M2 beads and immunoblotted with anti-HA. **b** Domain organization of TOLLIP protein. **c** Co-immunoprecipitation and immunoblot analysis of 293 T cells expressing TOLLIP mutants along with HA-ACE2. **d** Mass spectrometry analysis of a K48-linked ubiquitin peptide of purified ACE2. **e** 293 T cells transfected with plasmids expressing Flag-ACE2 and HA-K48-linked ubiquitin were treated with indicated concentration of 2-D08 for 24 h. The lysates were subjected to immunoprecipitation and immunoblot analysis. **f** Calu-3 cells were transfected with scrambled or *PIAS4*-specific siRNAs. The lysates harvested after Baf A1 (0.2 µM) treatment for 6 h, were immunoprecipitated and analyzed by immunoblotting using K48-linkage specific polyubiquitin antibody and anti-ACE2 antibody. **g** Immunoprecipitation and immunoblot analysis of 293 T cells expressing Flag-ACE2 (WT or K187R) and HA-K48-linked ubiquitin in the presence of Myc-PIAS4. **h** Mass spectrometry analysis identified a ubiquitination site on ACE2 at K187. **i** 293 T cells expressing HA-TOLLIP and wild-type (WT) Flag-ACE2 or its K187R mutant, followed by immunoprecipitation with anti-Flag beads and immunoblot analysis with anti-HA. **j** 293 T cells were transfected with WT Flag-ACE2 or its K187R mutant and treated with 2-D08 for indicated concentrations. The protein levels of Flag-ACE2 were analyzed by immunoblot. **k** Quantification of the protein levels in **j**. In **k**, all error bars, mean values ± SEM, *P*-values are determined by unpaired two-tailed Student's *t*-test of *n* = 3 independent biological experiments. For **a**, **c**, **e**–**g**, **i**, and **j**, similar results are obtained by three independent biological experiments. Source data are provided as a Source data file.

## Generation of stable expression cell lines

For the constitutive expression of ACE2, HEK293T cells transfected with FG-EH-DEST-ACE2-Puro, VSGV and Δ8.9 for 48 h were used to produce lentiviral particles. The lentivirus-containing supernatant was filtered through a 0.45 mm filter. After incubation with lentivirus-containing supernatant for 48 h together with Polybrene (8 µg ml⁻¹) [Santa Cruz Biotechnology, cat. sc-134220], ACE2 overexpressing A549 cells were selected with puromycin (Gibco, cat. A1113802). For *SENP* KO cells, pLentiCRISPRv2 expressing small guide RNA (sgRNA) were generated according to standard molecular biology technique and the sequences of target genes were described in Supplementary Table 4.

## Cytotoxicity assay

The supernatants of cells with indicated treatments were harvested for examining cell death by a lactate dehydrogenase (LDH) assay using CytoTox 96 Non-Radioactive Cytotoxicity kit (Promega, cat. G1780).

## Real-time quantitative PCR (qPCR)

Total RNA was extracted from cells using the TRIzol reagent (Invitrogen, cat. 15596018) and cDNA was generated with HiScript® II Q RT SuperMix for qPCR (+gDNA wiper) (Vazyme, cat. R223-01). qPCR analysis was performed using the 2 × RealStar Green Power Mixture (GenStar, cat. A311-10) and the data were normalized to *RPL13A* expression. The primer pair sequences could be found in Supplementary Table 5.

## Immunoprecipitation and immunoblot analysis

The whole-cell extracts collected from cells with indicated treatments were incubated with appropriate antibodies plus Protein A/G beads (Thermo Scientific, cat. 20424) at 4 °C for overnight. For immunoprecipitation with anti-Flag, α-FLAG M2 beads (Millipore, cat. A2220) were used. The beads were washed five5 times with low-salt lysis buffer (50 mM HEPES, 150 mM NaCl, 1 mM EDTA, 10% glycerol, 1.5 mM MgCl₂, and 1% Triton X-100) and eluted with 2 × SDS Loading Buffer (FD Biotechnology, cat. FD006) for immunoblotting. Proteins were transferred to PVDF membrane (Bio-Rad, cat. 1620177) and incubated with indicated antibodies. Protein bands were imaged with Immobilon Western Chemiluminescent HRP Substrate (Millipore, cat. WBKLS) in a ChemiDoc XRS + System (Bio-Rad Laboratories) using the Image Lab software.

## Mass spectrometry analysis

Total lysates prepared from 293 T cells transfected with plasmid encoding Flag-ACE2 were immunoprecipitated with α-FLAG M2 beads. The eluates were resolved by SDS-PAGE. and then stained with Coomassie Blue to excise the entire lane. Afterward, the samples were digested with trypsin and the peptides were chromatographed through the Easy-nLC 1000 system (Thermo Fisher) at Applied Protein Technology Corporation (Shanghai, China). LC-MS/MS identification of peptide mixtures was performed once with two replicates per condition using Q Exactive mass spectrometer (Thermo Fisher). RAW files generated by the spectrometer was processed in Proteome Discoverer (version 1.4) software with searching the library of Uniprot homo sapiens (uniprot_Homo_sapiens_188433_20200217.fasta) for protein identification. Trypsin was specified as the proteolytic enzyme, with up to two missed cleavage sites allowed. Carbamidomethylation was set as the fixed modification. Oxidation (M) and ubiquitination [GlyGly (K)] were set as the variable modifications. The precursor mass tolerance was set to 20 ppm and the fragment ion tolerance at 0.1 Da. Proteins were identified based on at least one unique peptide. Protein and peptide identification confidence threshold were set to an FDR of 1%. The tandem mass spectra of matched ubiquitinated peptides were further checked manually for their validity.

## SUMOylation assay and prediction of SUMOylation sites

For SUMOylation assays, cell lysed in low-salt lysis buffer supplemented with deSUMOylation inhibitor NEM (20 mM). Cell extracts sonicated for 20 s then clarified at 12,000 × *g* for 5 min and incubated with α-FLAG M2 beads at 4 °C for overnight. The samples were washed with low-salt lysis buffer for four times and the eluates were boiled at 100 °C for 5 min in 15 µl 2 × SDS Loading Buffer. GPS-SUMO, SUMOplot analysis (Abgent) software, Joined advanced SUMOylation site and SIM analyser (JASSA) and Ron Hay's SUMO consensus motif search tool (https://www.lifesci.dundee.ac.uk/groups/ron_hay/pages/SumomotifQuery.html) were used for the prediction of putative SUMOylation sites in ACE2.

## Confocal microscope

Cells cultured on glass-bottom culture dishes (Nest Scientific) were fixed with 4% paraformaldehyde for 10 min and permeabilized in methyl alcohol for 30 min at −20 °C. The samples washed with PBS for three times were blocked in 5% goat serum for 1 h and incubated with primary antibodies diluted in 5% goat serum for overnight. After washing with PBS for three times, the cells fluorescently labeled with secondary antibody for 1 h. Confocal images were detected using a super-resolution confocal microscope (TCS SP8 STED 3X; Leica) equipped with ×100 1.40 NA oil objectives and processed for gamma adjustments using Leica AS Lite or ImageJ software (National Institutes of Health). The data plotted on the line intensity plots were generated using the Plot Profile function of ImageJ on a single plane z-stack of confocal images. The colocalization was analyzed with no less than 10 cells per sample.

## SARS-CoV-2 virus stock production and infection

The SARS-CoV-2 strain was isolated from COVID-19 patients in Guangzhou (GenBank accession no. MT123290) and passaged on Vero E6 cells. Centrifuge the cell culture supernatant at 450 × *g* for 5 min at 4 °C to harvest SARS-CoV-2. Viruses were tittered by plaque assay and stored at −80 °C. Cells were infected the indicated MOI of SARS-CoV-2 as previously described[69,70]. For better adsorbing, the infected cells were gently rocked every 15 min at 37 °C for 1 hr before incubation at 37 °C for indicated lengths of time.

### Infection assay using SARS-CoV-2 S pseudotyped virus

Spike-pseudotyped virus-like particles (VLPs) were obtained from the HIV-1 lentiviral packaging system. Briefly, psPAX2, pHIV-Luciferase, and SARS-CoV-2 S protein expressing pcDNA3.1 vector (all these three plasmids are gifts from Yi-Ping Li laboratory) are co-transfected into the Vero E6 cells with mass ratio at 1:1:2 for 48 h. VLPs were harvested from culture medium by centrifugation at $200 \times g$ for 10 min and a second centrifugation at $2000 \times g$ for 10 min at 4 °C. The collected supernatant was filtered through a 0.45 μm filter membrane and centrifuged with 20% sucrose at $21,000 \times g$ for 7 h at 4 °C. The pellets were resuspended in ddH$_2$O. For infection assays, adherent ($1 \times 10^5$) cells plated in 24-well plates were infected with Spike-pseudotyped VLPs for 1–2 h at 37 °C for attachment and cultured for 48 h. The infection of SARS-CoV-2 S pseudotyped virus was detected by measuring luciferase activity (relative light unit; RLU) in cell lysates using Luciferase Substrate Buffer (Promega) with a Luminoskan Ascent luminometer (Thermo Scientific).

### SARS-CoV-2 plaque assay

The supernatants of virus-infected cells or mice lung homogenate were serially diluted in DMEM medium. Vero E6 cells cultured in 12-well plates were inoculated with the DMEM medium and gently rocked every 15 min at 37 °C for 1 h. Afterward, the inocula were removed and the plates were overlaid with 1.2% agarose containing 4% FBS. The overlays were further removed after incubation for 2 days, and the plaques were determined by staining with 0.1% crystal violet. Viral titers were calculated as PFU per gram tissue.

### Subcellular fractionations assays

Cells cultured in 10 cm culture dish were washed two times with PBS and collected using a cell scraper. After centrifugation at $600 \times g$ for 5 min at 4 °C, the cytosolic and membrane fractions of the harvested samples were further isolated using Mem-PER™ Plus membrane protein extraction kit (Thermo Scientific, cat. 89842).

### Immunofluorescence staining

Paraffin-embedded sections of lung biopsies were deparaffinized and incubated with blocking buffer (PBS with 5% goat serum and 0.3% Triton X-100) at room temperature for 1 h. The samples were then stained with primary antibodies in a wet chamber at 4 °C in the dark for overnight. After washing with PBS for three times, the sections were incubated with secondary antibodies at room temperature in the dark for 1 h, and were stained with DAPI for nuclear quantitation. Immunofluorescent images were obtained with a Nikon Eclipse Ni-E microscope using a ×40 objective. The protein expression levels were quantified in square micrometers by using ImageJ software.

### RNA-seq analysis

Total RNA was extracted from cells using TRIzol reagent, and sequencing was conducted by Sangon Biotech. Bioanalyzer (Agilent 2100 Bioanalyzer) was used to assess the quality of samples. RNA-seq libraries of polyadenylated RNA were generated using mRNA-seq V2 Library Prep Kit and sequenced on MGISEQ-2000 platform. All clean data were mapped to the human genome GRCh38 using HISAT2 v2.1.0 with default parameters and BAM files were sorted by Samtools 1.9. Reads counts were summarized using the feature-Counts program as part of the Subread package release 2.0.0 (http://subread.sourceforge.net/)[71]. DESeq2 was used to normalize reads counts and $p$-value < 0.05 and absolute logged fold-change ≥ 1 were determined as differentially expressed genes (DEGs). Bioconductor clusterProfiler package v3.14.3 was applied to analyze the functional enrichment of DEGs[72,73]. RNA-seq sequence density profiles were normalized using bamCoverage and visualized in IGV genome browser[74]. The full gene lists of our RNA-Seq experiments were provided in the Supplementary Data 2 and 3.

### Statistical analysis

Data from three or more experiments were displayed as mean ± SEM unless otherwise indicated, and data of immunofluorescent staining experiments are represented as mean ± SD of indicated image numbers. Student's $t$-test was used for all statistical analyses with the GraphPad Prism 8 software. Differences between two groups were considered significant when $P$-value was less than 0.05.

### Reporting summary

Further information on research design is available in the Nature Research Reporting Summary linked to this article.

## Data availability

The mass spectrometry proteomics data have been deposited to the ProteomeXchange Consortium via the PRIDE partner repository with the dataset identifier PXD034168. RNA-seq data have been submitted to the Gene Expression Omnibus (GEO) under accession number GSE171130. The whole-genome sequence (WGS) of SARS-CoV-2 isolate used in this study can be found under GenBank accession number MT123290. The data that support the findings of this study are available from the corresponding authors upon request. Source data are provided with this paper.

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

## Acknowledgements

This work was supported by the National Key R&D Program of China (2020YFA0908700 to J.C., 2020YFC0842400 to J.Z.), National Natural Science Foundation of China (32170876 and 31970700 to S.J., 92042303 and 31870862 to J.C.), Guangdong Basic and Applied Basic Research Foundation (2020B1515120090 to J.C.), Natural Science Foundation of Guangdong Province, China (2021A1515012179 to Y.Wu), and the Fundamental Research Funds for the Central Universities, Sun Yat-sen University (22qntd2601 to W.X.).

## Author contributions

J.C. and S.J. conceived the project and designed the experiments. S.J. and X.H. performed the experiments. L.M., Z.Zhuang, Y.Wang, M.L., S.C., L.W., Z.W., Z.Zhao, Y.Wu, L.S., C.L., W.X., Y.Z., Z.S, K.P., and J.Z. provided technical help. S.J., L.M., and X.H. analyzed the data. J.C. provided resources, and directed the research. S.J. and J.C. wrote the manuscript. All authors read and approved the final manuscript.

## Competing interests

The authors declare no competing interests.
