## [Peer Review File · Nature Communications]

REVIEWER COMMENTS

Reviewer #1 (Remarks to the Author):

In the manuscript 'Suppression of ACE2 SUMOylation protects against SARS-CoV-2 infection through TOLLIP-mediated selective autophagy', Jin and colleagues describe a novel mechanism of ACE2 regulation that involves SUMOylation and ubiquitination of ACE2. The authors demonstrate that ACE2 can be modified by SUMO3 at K187, and that this is regulated by SENP3 and PIAS4. Furthermore, inhibition of SUMOylation at K187 leads to ubiquitination at this lysine residue, and degradation of ACE2 in a TOLLIP- and autophagy-dependent fashion.

The authors also try to link the stability of ACE2 to SARS-CoV-2 infection. In my opinion, the biggest flaw of the manuscript is that the authors claim that regulation of ACE2 by SUMO affects SARS-CoV-2 entry, without directly assaying viral entry. The infection assays are not designed in a way that would inform about viral entry directly. Instead, the experimental readout is also affected by later stages of viral life cycle (i.e. viral RNA transcription, translation or virus particle production). SARS-CoV-2 replication and virus production of course will be reduced if viral entry is reduced. Currently, the way the manuscript is written suggests that the performed experiments would allow for conclusions on virus entry. This needs to be addressed and the authors need to clarify that the presented data does not inform about virus entry directly and instead provides information about the entire viral life cycle. In my opinion, the quality of the manuscript could be improved significantly if the authors were able to add experimental data to show that ACE2 SUMOylation affects SARS-CoV-2 entry, for example using Spike-pseudotyped virus-like particles (as done in Fig 3e). I would suggest repeating key experiment using this system to demonstrate a correlation of SUMO-mediated regulation of ACE2 and SARS-CoV-2 entry.

I would consider the manuscript suitable for publication in Nature communications if the authors can provide data showing that the presented mechanism of ACE2 regulation by SUMOylation has an impact on SARS-CoV-2 entry.

Major points:

- Figure 1a: The authors use a CoIP-MS approach to identify proteins that interact with ACE2. Could the authors please clarify which was the control IP condition used in this experiment (e.g. IP using IgG, or IP from cells that express the tag only), and how this information about unspecific background proteins was incorporated in the data analysis. Without this control it is difficult to draw conclusions about specific interactors of ACE2 from this experiment.

- Figure 2: Here the authors treat Calu-3 cells or hACE2 transduced mice with inhibitors that globally inhibit SUMOylation. As a read-out, the authors use transcriptome analysis of infected cells/tissues and viral titers or viral RNA load. In line 147/148 the authors conclude that their results demonstrate that suppressing ACE2 SUMOylation inhibits the entry of SARS-CoV-2. There are three major issues with this interpretation of the data.

a) This assay is not specific for ACE2 SUMOylation. The inhibitors used here affect SUMOylation of all SUMO targets in the cell. Thus, the text should be revised accordingly (e.g. line 125 to 128, line 147/148). The interpretation of this data is very difficult, since SUMOylation is involved in regulating many cellular processes, including transcription, stress responses and the IFN pathway. Thus, global inhibition of SUMOylation can affect the regulation of the virus life cycle as well as the host response on many different levels. In addition, viral proteins could potentially undergo SUMOylation during the viral life cycle, and global inhibition of SUMOylation could thus directly affect functions of viral proteins, and thereby viral replication.

b) The read-out of the infection assay is at viral RNA level or production of infectious virus. Thus, one cannot conclude that virus entry is affected – it might well be later stages of the viral life cycle.

c) It is not clear why the authors chose a transcriptome analysis as a read out here. As mentioned above, this is difficult to interpret, since SUMO is involved in regulation of transcription and in regulation of antiviral/inflammatory responses. The reduced host response signature in the presence of SUMO inhibitors could be caused by reduced viral replication in the cell, or by a more direct effect of the global inhibition of SUMOylation.

I think the authors should revise the conclusions that can be drawn from this assay, and discuss the different aspects of how the global inhibition of SUMOylation could affect virus replication and the host response to viral infection.

In order to draw conclusions about SARS-CoV-2 entry, it is necessary to perform additional experiments that are designed to analyze viral entry, e.g. using Spike-pseudotyped VLPs. Using this assay in combinations with SUMO inhibitors allows for conclusions about how SUMOylation (in general) affects viral entry. Additional experiments using a specific viral entry assay should be performed to correlate the SUMO-mediated regulation of ACE2 with SARS-CoV-2 entry. This could for example be achieved by using knockdown or knockout systems to deplete PIAS4 or TOLLIP.

- Figure 3g,i: Here the authors observe SARS-CoV-2 N protein in a western blot, which is not a read out for entry, but for viral protein expression. Please revise the conclusion in the text accordingly (lines 176, 186 and 189)

- Fig. 5b,c,d,e: The authors show increased co-localization of ACE2 and TOLLIP over the course of SARS-CoV-2 infection, and increased interaction of ACE2 with PIAS4 and SENP3. However, this interaction occurs after the virus has already entered the cells, and thus cannot affect virus entry into the cell and might not be of physiological relevance for infection. Please include a discussion of this aspect in the manuscript.

- Fig. 5g,h,i. The authors knock down TOLLIP in Calu-3 cells and perform an RNA-Seq experiment in infected cells. It is not clear to me why the author chose a transcriptome analysis here. They conclude that TOLLIP deficiency potentiated the cellular immune response to infection (line 271-273). However, in my opinion the increased antiviral response in the absence of TOLLIP is likely caused by the increased viral replication in these cells, and not as implied by the author, by a direct regulation of the innate immune response by TOLLIP. Please include a discussion of these aspects in the manuscript.

Additionally, this experiment has only been performed using one siRNA for TOLLIP, and it would be advisable to include a second siRNA targeting TOLLIP to exclude potential off target effects.

- Another question that comes to mind is if solely the stability of ACE2 affects viral entry, or if SUMOylation (or ubiquitination) of ACE2 could also impact the ability of SARS-CoV-2 spike to bind to ACE2. If not addressed experimentally, this should at least be discussed based on available data about molecular requirements of ACE2-Spike interactions.

- In this context it is also critical to discuss the localization of the SUMOylated residue K187 of ACE2. From the current models, I understand it is believed that amino acid 1-740 are located on the extracellular domain of ACE2. How do the authors consolidate this with their model of SUMO/PIAS4/SEN3/TOLLIP-mediated regulation of ACE2? In other words, if the SUMOylated residue is in the extracellular domain of ACE2, does this mean that SUMOylation/deSUMOylation of ACE2 happen in the ER/Golgi pathway?

Minor points:

- Figure 1a: Does the color code state the number of identified proteins belonging to the respective GO term? Please state in the figure legend.

- The description for the expression constructs of ACE2 K-to-R mutants shown in Fig.1i is missing in the methods part.

- Extended Fig. 1b: Could you please state how many cells were analyzed here? Please add a description of the co-localization analysis to the methods part.

- Fig. 1g/Ext. Fig 1b: SUMO3 is reported to mainly localize to the nucleus, but here a strong plasma membrane staining is observed. Could the authors offer an explanation for this discrepancy? Additionally, I believe that this is an analysis of SUMO2/3, and not SUMO3 as stated, since the paralogues cannot be distinguished by antibody detection. Furthermore, the description of the confocal microscope is missing in the methods part.

- line 115-117: 'cells treated with ML-792 ... exhibited dramatically lower appearance of SUMOylated ACE2' – this should be corrected to 'lower co-localization of SUMO2/3 and ACE2' since the microscopy assay does only allow for conclusions about localization of ACE2 and SUMO, not about actual SUMO-conjugation to ACE2.

- I think the authors should make the full gene list from RNA-Seq experiments available to enable the reader to recapitulate the presented GO and GSEA analyses (data used for Fig. 2 and Fig. 5)

- Fig 2: I suggest to add experiments to demonstrate that the inhibitor treatment used in this experiment does not affect viability of the Calu-3 cells, since this could also lead to impaired virus replication.

- For Figure 2 and 5, the authors state in the figure legend that these are microarray data. However, in the main text and the method part they talk about RNA seq analysis. Please clarify.

- The siRNA transfection experiments in Fig. 3g,h,i as well as in Fig. 5e,f,g,h and Ext.Fig 5g are only done with one siRNA per gene. It would be useful to repeat the assays with a second siRNA, to exclude that the observed phenotypes are caused by potential off-target effects of the siRNA.

- line 192-196 / Fig. 4a: It looks like ACE2 is upregulated in the presence of the inhibitors and does not further increase by overexpression of PIAS4, rather than the PIAS4-mediated upregulation being blocked as stated by the authors. Adding a summary quantification analysis of the western blot signals to visualize the results of the three independent experiments could help the reader to judge the data.

- In general, it would be useful to add a quantification analysis of western blot signals obtained in independent experiments for all western blots where the differences are difficult to judge from the picture of only one membrane (e.g. Fig. 4b,d,e; Ext.Fig. 5e,f; Fig.6a)

- line 198: It should be explained how Rapamycin and EBSS work in the cell

- Fig. 4i: Could you explain why the fractionation is performed in the presence of 3-MA? How does this affect the results?

- Fig. 4h: What does 'Group 1', 'Group 2', 'Group 3' mean?

- Fig 5b: Showing in addition a higher magnification (zoom in) of the microscopy pictures would help the reader to judge the images.

- Ext. Fig 4g: A description of the ACE2 Δ CT+TM mutant is missing. It is not clear, which parts of the protein have been deleted in this mutant. It would also be useful to cite respective literature where this mutant was described as I believe it is supposed to act as a control in this assay.

- line 306/307: 'K48-linked ubiquitination of ACE2 was enhanced in SENP3-deficient cells' – 'ubiquitination' should be replaced by 'co-localization' since this is an IF assay that does not allow for conclusion about covalent ubiquitination.

- The following experimental details are missing in the methods section:

o Infection assay using Spike-pseudotyped VLPs

- o SARS-CoV-2 virus stock production
- o Description of many of the used expression plasmids

Reviewer #2 (Remarks to the Author):

Suppression of ACE2 SUMOylation protects against SARS-CoV-2 infection through TOLLIP-mediated selective autophagy.

In this well-written manuscript, the authors explored how endogenous protein quality control measures affect the abundance of ACE2, the main receptor used by the pandemic SARS-CoV-2 for cellular entry, and how this affects virus-triggered inflammatory responses in mouse lung tissue. The overall concept presents novel data on ACE2 post-translational modification, its alteration in response to SARS-CoV-2 infection. The authors elegantly demonstrate SUMO3 conjugation of ACE2, leading to its stabilization due to less ubiquitination, the latter being required to mark ACE2 for recognition by TOLLIP, directing it to autophagic degradation. The authors used a broad spectrum of state-of-the-art methodology, including NGS RNA seq, proteomics, bioinformatics, SARS-CoV-2 virology and infection experiments in mice (upon Ad transfer of the huACE2 cDNA), very elegant protein biochemistry and complementary approaches that strengthen the respective data. Very inhibitory compounds were used for cell culture experiments and altogether these experiments documented K48-linked ubiquitination of ACE2 by less SUMOylation as a putative strategy to mitigate SARS-CoV-2 infection.

The biochemical documentation of SUMOylation of ACE2 and the contribution of PIAS4 and the de-conjugating enzyme SENP3 is convincing. In addition, the experiments illustrating ACE2-ubiquitination and TOLLIP-dependent allocation to autophagy provided strong evidence for the respective conclusions.

Major:

Although the authors managed to generate some in vivo evidence that is indeed supportive for the role of SUMOylation on SARS-CoV-2 infection, this part of the manuscript does not provide sophisticated evidence on the proposed link of SUMOylation, ACE2 abundance and SARS-CoV-2 infection. The manuscript would strongly improve if conclusive data on SUMOylation-dependent ACE2 expression locally in infected cells could be obtained. The proposed mouse model that relies on Ad5-mediated hACE2 expression and the systemic application of SUMOylation inhibitors is questionable regarding its clinical relevance, but also regarding its suitability in the context of the paper. Are both conjugation processes, SUMO+ubi, as conserved as needed to conclude on ACE-2 abundance? Is SUMOylation indeed affected by the respective inhibitors, how is ACE-2 in AEC controlled by these aspects, is indeed the viral entry affected or the replication process affected by alternative SUMOylation-dependent processes. Coronaviruses, at least partially, utilize the

autophagic pathway for replication. In addition the demonstration/elucidation of the respective pathways in lung tissue, the proposed effects on the SARS-CoV-2 triggered immune response deserve further detailed demonstration, e.g. of lung pathology, CT scans, BAL etc. to conclude part is elegant, but less well documented in terms of viral-infection triggered immune responses.

More data is required on the role of SUMOylation on SARS-CoV-2 replication. Fig 5f is a linear or Log10 scale? There is a lot of indirect evidence for the proposed mechanisms that affect SARS-CoV-2 entry. The authors should explain in detail which experiments precisely document virus entry as opposed to altered replication and (thereby influenced) immune/cellular responses.

Minor:

Several typos and grammar issues need to be addressed.

Since many different inhibitors are used, it is recommended to provide a table, listing the compounds and their respective targets.

Reviewer #3 (Remarks to the Author):

The respected authors provided substantial amount of findings about role of TOLLIP-mediated selective autophagy in Suppression of ACE2 SUMOylation and its effect in SARS-CoV-2 infection. Although the data is interesting and convincing in some aspect of their studies but the following concepts should be considered.

1- The first important point is the effects and concentrations of Bafilomycin (200 nM, 6 hrs) and 3-MA (10 mM 6 hrs) and Rapamycin (250 nM). These concentration of inhibitors could be extremely toxic for some cell lines. It is important the respected authors provide evidence about the viability of cells in all conditions.

2- It is strongly recommended that the respected authors provide blots in each conditions for LC3 lipidation, p62 degradation, Beclin-1 and Atg5-12 conjugations.

3- Based on the data that has been presented, I believe beyond selective autophagy, we may have effects of autophagy in the intracellular trafficking and its effect of ACE2. I recommend the respected authors provide evidence for using different concentration of Baf-A1 (0.1, 1, 2.5, 10 nM) and co-treat them with the SARS-CoV-2 during all infection time and look at ACE2 expression and

also localize ACE-2 with early and late endosomes, and early late exosomes in different Baf-A1 concentrations. It may give us a much better understanding of the role of autophagy in ACE-2 life cycle in the infected cells.

4- We all know the secretory autophagy is playing a substantial role in cellular immune response of the infected cells. It is important that the respected authors provide evidence about the role of secretory autophagy in regulation of cytokine/chemokine profile of the infected cells using 2 low concentration of Baf-A1. It may have effect in ACE2 recycling process.

5- The data for the Cav-1 is very interesting. It directs us the potential role of SARS-CoV-2 lipophagy and indirect effect of lipid raft. It is recommended the respected authors provide evidence about the potential role of SARS-CoV-2 infection in lipophagy and also use lipidomics to measure different contents of lipids like cholesterol and ceramides that are important for lipid raft and cytosolic membranes.

Dear Referees,

Thank you very much for providing us a valuable opportunity to revise our paper entitled “Suppression of ACE2 SUMOylation protects against SARS-CoV-2 infection through TOLLIP-mediated selective autophagy”. We have performed additional experiments and provided new data in accordance with the Referees’ suggestions. These results strengthen our overall conclusions and profoundly improve the manuscript. A point-by-point response to the Referees’ concerns is included below. For the convenience, we have numbered all the suggestions and pasted all the figures in response to each question in this letter.

We hope that the revised manuscript meets the requirements for publication in “*Nature Communications*”, and we look forward to hearing from you.

Sincerely,

Jun Cui, Ph.D.

Professor and Dean of Department of Biochemistry

School of Life Science

Sun Yat-sen University

(The corresponding author)

Response to the comments of Reviewer #1

1. Figure 1a: The authors use a CoIP-MS approach to identify proteins that interact with ACE2. Could the authors please clarify which was the control IP condition used in this experiment (e.g. IP using IgG, or IP from cells that express the tag only), and how this information about unspecific background proteins was incorporated in the data analysis. Without this control it is difficult to draw conclusions about specific interactors of ACE2 from this experiment.

Response: We thank the reviewer for this remark. Lysates prepared from HEK 293T cells transfected with plasmid encoding ACE2 (with Flag tag in the C terminal) or Flag-tagged empty vector as negative control were subjected to immunoprecipitation with anti-Flag antibody. The precipitated proteins were resolved by SDS-PAGE and subjected to mass spectrometry (MS) analysis after in-gel digestion with trypsin. This experiment was performed once with two replicates per condition. To exclude the unspecific background proteins, the results of MS data set of Flag-tagged empty vector group were got rid of the ACE2 group. We have added this description in the method part and figure legend.

2. Figure 2: Here the authors treat Calu-3 cells or hACE2 transduced mice with inhibitors that globally inhibit SUMOylation. As a read-out, the authors use transcriptome analysis of infected cells/tissues and viral titers or viral RNA load. In line 147/148 the authors conclude that their results demonstrate that suppressing ACE2 SUMOylation inhibits the entry of SARS-CoV-2. There are three major issues with this interpretation of the data.

a) This assay is not specific for ACE2 SUMOylation. The inhibitors used here affect SUMOylation of all SUMO targets in the cell. Thus, the text should be revised accordingly (e.g. line 125 to 128, line 147/148). The interpretation of this data is very difficult, since SUMOylation is involved in regulating many cellular processes, including transcription, stress responses and the IFN pathway. Thus, global inhibition of SUMOylation can affect the regulation of the virus life cycle as well as the host response on many different levels. In addition, viral proteins could potentially

undergo SUMOylation during the viral life cycle, and global inhibition of SUMOylation could thus directly affect functions of viral proteins, and thereby viral replication.

b) The read-out of the infection assay is at viral RNA level or production of infectious virus. Thus, one cannot conclude that virus entry is affected – it might well be later stages of the viral life cycle.

c) It is not clear why the authors chose a transcriptome analysis as a read out here. As mentioned above, this is difficult to interpret, since SUMO is involved in regulation of transcription and in regulation of antiviral/inflammatory responses. The reduced host response signature in the presence of SUMO inhibitors could be caused by reduced viral replication in the cell, or by a more direct effect of the global inhibition of SUMOylation.

I think the authors should revise the conclusions that can be drawn from this assay, and discuss the different aspects of how the global inhibition of SUMOylation could affect virus replication and the host response to viral infection.

In order to draw conclusions about SARS-CoV-2 entry, it is necessary to perform additional experiments that are designed to analyze viral entry, e.g. using Spike-pseudotyped VLPs. Using this assay in combinations with SUMO inhibitors allows for conclusions about how SUMOylation (in general) affects viral entry. Additional experiments using a specific viral entry assay should be performed to correlate the SUMO-mediated regulation of ACE2 with SARS-CoV-2 entry. This could for example be achieved by using knockdown or knockout systems to deplete PIAS4 or TOLLIP.

Response: We thank the reviewer for this comment and agree with his/her thoughtful opinion. SUMOylation inhibitors may affect SUMOylation of all cellular SUMO targets and more specific inhibitor of ACE2 SUMOylation will be crucial for the intensive study of ACE2 and SARS-CoV-2 infection, however, there are no commercial inhibitor of specific E3 SUMO ligase or deSUMOylation enzyme. Instead, we knocked down the expression of *PIAS4* and *SEN3* as the reviewer suggested, observing that *PIAS4* depletion decreased the amount of cellular

pseudotyped SARS-CoV-2, while *SENP3* knockdown increased the pseudotyped SARS-CoV-2 infection (**New Fig. 1a** and **1b**, related to Fig. 3f,i in the revised manuscript). Since the SARS-CoV-2 S pseudotyped virus only contains the spike protein of SARS-CoV-2^{1,2}, this assay may reduce the possibility that SUMOylation of ACE2 affects the SARS-CoV-2 infection caused by the SUMOylation of viral protein or key host immune proteins.

We agree with the reviewer that detection of viral RNA and production of infectious virus may be not sufficient to support the viral entry. The reason why we chosen transcriptome analysis here is that we want to provide some insights into the potential SARS-CoV-2 entry-induced immune response. As this is an indirect assay to reflect the entry of SARS-CoV-2, we applied the SARS-CoV-2 S pseudotyped virus for more experiments and found that treatment of SUMOylation inhibitors decreased the infection of cellular pseudotyped SARS-CoV-2 and the levels of SARS-CoV-2 S pseudotyped virus were increased in *TOLLIP*-deficient cells (**New Fig. 1c** and **1d**, related to Fig. 2c and Fig. 5e in the revised manuscript). Combining the results that reversible SUMOylation of ACE2 affects the its stabilization, and its essential role in viral entry, we concluded that SUMO modification of ACE2 modulates the SARS-CoV-2 entry in the pseudotyped SARS-CoV-2 system. Despite the additional results indicated that SUMOylation of ACE2 may influence the entry of pseudotyped SARS-CoV-2, it is difficult to study the entry of real SARS-CoV-2, so we have changed “the viral entry of SARS-CoV-2” into “SARS-CoV-2 infection” in the manuscript as the reviewer suggested.

New Figure 1. (a) Luciferase activity of Calu-3 cells transfected with scramble or *PIAS4*-specific siRNAs, along with SARS-CoV-2 S pseudotyped virus infection for 48 hr. (b) Calu-3 cells transfected with scramble or *SENP3*-specific siRNAs were infected with SARS-CoV-2 S pseudotyped virus for 48 hr, the protein extracts were harvested for luciferase assay. (c) Luciferase activity of Calu-3 cells treated with 2-D08 (200 μ M), ML-792 (10 μ M), GA (5 μ M), or TAK-981 (5 μ M), along with SARS-CoV-2 S pseudotyped virus infection for 48 hr. (d) Calu-3 cells transfected with scramble or *TOLLIP*-specific siRNAs were infected with SARS-CoV-2 S pseudotyped virus for 48 hr, the protein extracts were harvested for luciferase assay. Data in (a–d) are expressed as means \pm SEM of three independent experiments. *** p < 0.001 (two-tailed Student’s t -test).

3. Figure 3g,i: Here the authors observe SARS-CoV-2 N protein in a western blot, which is not a read out for entry, but for viral protein expression. Please revise the conclusion in the text accordingly (lines 176, 186 and 189)

Response: We have revised our conclusion as the reviewer suggested and the modified text has been highlighted with yellow.

4. Fig. 5b,c,d,e: The authors show increased co-localization of ACE2 and TOLLIP over the course of SARS-CoV-2 infection, and increased interaction of ACE2 with

PIAS4 and SENP3. However, this interaction occurs after the virus has already entered the cells, and thus cannot affect virus entry into the cell and might not be of physiological relevance for infection. Please include a discussion of this aspect in the manuscript.

Response: We appreciate the reviewer for this comment. Immunofluorescence and co-immunoprecipitation analysis indicated that the ACE2-TOLLIP association was increased upon SARS-CoV-2 (**Fig. 5b–d**). Intriguingly, we observed the association between ACE2 and SENP3 was enhanced while the interaction between ACE2 and PIAS4 was decreased upon SARS-CoV-2 infection (**Fig. 5d**). The association between ACE2 and TOLLIP/PIAS4/SENP3 could be detected under basal levels in the soluble cytosol (**New Fig. 4b**, related to Supplementary Fig. 3j). We think the SUMOylation system controls the protein level of ACE2 at the basal conditions to further influence the subsequent SARS-COV-2 infection. We strongly agree with the reviewer that this changed interaction occurs after the virus has already entered the cells, and thus cannot affect virus entry into the cell and might not be of physiological relevance for infection. There may be a competitive interaction of PIAS4 and SENP3 with ACE2. We guess the post-translation modification or the upstream activation signaling lead to the altered association between SENP3 and ACE2 under SARS-CoV-2 infection. mTOR-mediated phosphorylation of serine/threonine residues within the N-terminal domain in SENP3 and the inhibition of mTOR trigger the nucleolar release of SENP3³. Cyclin B-dependent kinases 1 and protein phosphatase 1 α were identified as the kinase and phosphatase in control of mitotic SENP3 phosphorylation, respectively⁴. In response to DNA damage, p53 suppresses SENP3 phosphorylation to mediate G2 checkpoint⁵. Since SARS-CoV-2 infection causes the dysregulation of mTOR signaling and mitosis^{6,7}, we hypothesize the phosphorylation of SENP3 could be further influenced during SARS-CoV-2 infection, thereby affecting the association between ACE2 and PIAS4/SENP3/TOLLIP. The involvement of SARS-CoV-2 infection in modulating ACE2 SUMOylation can be an important field for dissection in the future. We have added this part in the discussion as suggested by the reviewer.

5. Fig. 5g,h,i. The authors knock down TOLLIP in Calu-3 cells and perform an RNA-Seq experiment in infected cells. It is not clear to me why the author chose a transcriptome analysis here. They conclude that TOLLIP deficiency potentiated the cellular immune response to infection (line 271-273). However, in my opinion the increased antiviral response in the absence of TOLLIP is likely caused by the increased viral replication in these cells, and not as implied by the author, by a direct regulation of the innate immune response by TOLLIP. Please include a discussion of these aspects in the manuscript.

Additionally, this experiment has only been performed using one siRNA for TOLLIP, and it would be advisable to include a second siRNA targeting TOLLIP to exclude potential off target effects.

Response: The reason why we chose transcriptome analysis here is that we want to provide some insights into the potential SARS-CoV-2 entry induced immune response. As this is an indirect assay to reflect the entry of SARS-CoV-2, we applied the SARS-CoV-2 S pseudotyped virus for more experiments and observed the levels of SARS-CoV-2 S pseudotyped virus were increased in TOLLIP-deficient cells (**New Fig. 1d**, related to Fig. 5e in the revised manuscript). We agree with the reviewer that the increased antiviral response in the absence of TOLLIP might be caused by the increased viral replication in these cells. We have added this part in our discussion. Moreover, we employed two siRNAs for the study and obtained the similar results (**New Fig. 2**, **New Fig. 7c** and **7e**, related to Fig. 5f,g in the revised manuscript and Supplementary Fig. 5j).

New Figure 2. Plaque titration of SARS-CoV-2 in supernatants of scrambled or TOLLIP-specific siRNAs transfected Calu-3 cells infected with SARS-CoV-2 (MOI =

0.5). Data represented mean \pm SEM of triplicate experiments. *** $p < 0.001$ (two-tailed Student's t -test).

6. Another question that comes to mind is if solely the stability of ACE2 affects viral entry, or if SUMOylation (or ubiquitination) of ACE2 could also impact the ability of SARS-CoV-2 spike to bind to ACE2. If not addresses experimentally, this should at least be discussed based on available data about molecular requirements of ACE2-Spike interactions.

Response: To address the reviewer's concern, we checked the association between ACE2 and SARS-CoV-2 spike protein, observing that the interaction remains largely unchanged with SUMOylation inhibitor treatment (New Fig. 3a, related to Supplementary Fig. 5f). Moreover, we studied the recognition of SARS-CoV-2 Spike by ACE2 and found that the ACE2 K187R mutant, which SUMOylation and ubiquitination were abrogated, could still interact with the spike protein (New Fig. 3b, related to Supplementary Fig. 5g). These results suggested that SUMOylation and ubiquitination at K187 may not affect the ACE2-spike protein interaction. In our manuscript, we think solely the stability of ACE2 mediated by SUMOylation influences viral infection, however, we believe post-translation modifications of ACE2 at other amino acids may modulate the interaction between ACE2 and SARS-CoV-2 spike protein. We have added this in our discussion part as the reviewer suggested.

New Figure 3. (a) Flag-Spike of SARS-CoV-2 was expressed in 293T cells, purified using Flag affinity column, and eluted with Flag peptide. HA-ACE2 was purified from 293T cells transfected with plasmid encoding ACE2 (with HA tag in the C

terminal) in the absence or presence of 2-D08 (200 μ M). HA-ACE2 was purified using HA affinity column, and eluted with HA peptide. Purified HA-ACE2 was incubated with immunopurified Flag-Spike in reaction buffer *in vitro*. After pull-down with Flag-beads, the bound material was analyzed by western blot using the indicated antibodies. **(b)** Flag-Spike and HA-ACE2 as well as its indicated mutant were purified from 293T cells, and were incubated in reaction buffer *in vitro*. After pull-down with Flag-beads, the bound material was analyzed by western blot using the indicated antibodies. For data in **(a and b)**, similar results were obtained by three independent biological experiments.

7. In this context it is also critical to discuss the localization of the SUMOylated residue K187 of ACE2. From the current models, I understand it is believed that amino acid 1-740 are located on the extracellular domain of ACE2. How do the authors consolidate this with their model of SUMO/PIAS4/SEN3/TOLLIP-mediated regulation of ACE2? In other words, if the SUMOylated residue is in the extracellular domain of ACE2, does this mean that SUMOylation/deSUMOylation of ACE2 happen in the ER/Golgi pathway?

Response: We thank the reviewer for this question. As the reviewer pointed out, the aa 1–740 is traditionally thought as extracellular domain of ACE2. Recently, several research groups have demonstrated that aa 1–740 can undergo multilayer post-translation modifications. ACE2 is reported to be also localized at the cytoplasm when treated with metformin or AICAR, and AMP-activated protein kinase (AMPK) phosphorylates ACE2 at serine 680 for the protein stabilization of ACE2 in endothelial cells⁸. Casein kinase 1 alpha (CK1 α) phosphorylates ACE2 at serine 3/4/5 to trigger its recognition by the E3 ligase SPOP, and CK1/SPOP binding occurs in cytoplasm to facilitate ACE2 trafficking to plasma membrane⁹. Epigenetic eraser enzyme LSD1 and ACE2 colocalize both in the cytoplasm and nucleus. LSD1 targets ACE2 to remove the methylation of ACE2 at lysine 31. Intriguingly, ACE2 harbors a high-affinity, highly conserved nuclear localization sequence (NLS) within the cytoplasmic tail, thus ACE2 can be shuttled to the nucleus through interacting

importin- α^{10} . These results indicated that ACE2 distributes in the cytomembrane, cytoplasm and nucleus. To resolve the reviewer’s concern on our working model how ACE2 was SUMOylated, we enriched the integral membrane proteins and membrane-associated proteins from cultured Calu-3 cells, found that ACE2 could be displayed both in the membrane fraction and soluble cytosolic fraction. Next, we carried out coimmunoprecipitation assay to study the distribution of PIAS4/SEN3/TOLLIP and observed that PIAS4/SEN3/TOLLIP co-exists with ACE2 in the soluble cytosolic fraction (**New Fig. 4a**, related to Supplementary Fig. 3j). These results suggested that ACE2 undergoes SUMOylation and autophagic degradation in the cytoplasm, which is not dependent on ER/Golgi pathway.

ACE2 contains a putative N terminal signal peptide (SP, aa 1–17)¹¹. The presence of signal peptides facilitates N-tail translocation at the ER membrane, however, recent work showed that signal peptides of GPCRs do not only serve “classical” functions in the early secretory pathway. Uncleaved pseudo signal peptides may regulate receptor densities in the plasma membrane, receptor dimerization, and G protein coupling selectivity¹². We deleted the SP region of ACE2, and finding that the ACE2 Δ SP protein could still localize on the cell membrane (**New Fig. 4b**, related to Supplementary Fig. 3k). These results imply that besides the ER/Golgi pathway, there are other mechanisms that modulate the plasma membrane localization and cellular shuttle of ACE2.

New Figure 4. (a) Membrane and cytosolic fractions of Calu-3 cells were immunoblotted with antibodies directed against ACE2, a cytosol marker (α -tubulin), an ER membrane marker (SEC61 β), a Golgi apparatus membrane marker (Syntaxin 6), or a PM marker (E-cadherin). The membrane and cytosolic fractions were subjected to immunoprecipitation with anti-ACE2 and immunoblot analysis with

indicated antibodies. **(b)** Confocal microscopy of HeLa cells transfected with plasmids encoding WT and the Δ SP mutant form of ACE2, followed by labeling of ACE2 (green) with specific antibodies. Scale bar, 20 μ m. For data in **(a)**, similar results were obtained by three independent biological experiments.

8. *Figure 1a: Does the color code state the number of identified proteins belonging to the respective GO term? Please state in the figure legend.*

Response: We have used color code to clearly display the value of $(-\log_{10}\text{FDR})$ of the respective GO terms and have optimized our figure legend in the revised version.

9. *The description for the expression constructs of ACE2 K-to-R mutants shown in Fig. 1i is missing in the methods part.*

Response: We thank the reviewer for this remark and have added the description in our methods.

10. *Extended Fig. 1b: Could you please state how many cells were analyzed here? Please add a description of the co-localization analysis to the methods part.*

Response: We used 10 cells for statistical analysis and we have added the description in our methods as the reviewer suggested.

11. *Fig. 1g/Ext. Fig 1b: SUMO3 is reported to mainly localize to the nucleus, but here a strong plasma membrane staining is observed. Could the authors offer an explanation for this discrepancy? Additionally, I believe that this is an analysis of SUMO2/3, and not SUMO3 as stated, since the paralogues cannot be distinguished by antibody detection. Furthermore, the description of the confocal microscope is missing in the methods part.*

Response: We agree with the reviewer that SUMO3 mainly localizes in the nucleus. Since SUMO conjugation promotes the stabilization and cytomembrane distribution of ACE2, the strong plasma membrane staining of SUMO may be caused by the overexpression of ACE2. The coexistence of ACE2 and SUMOylation may lead to the

strong localization of SUMO3 at the cytomembrane (**New Fig. 5**). As mature SUMO2 and SUMO3 are virtually identical, it is hard to distinguish the paralogues by antibody detection. We have changed our label and modified the description in our text. Additionally, we have added the description of confocal microscope in our methods.

New Figure 5. Confocal microscopy of HeLa cells transfected with empty vector or plasmid encoding ACE2, followed by labeling of ACE2 (green) and SUMO2/3 (red) with specific antibodies. Scale bar, 20 μm .

12. line 115-117: ‘cells treated with ML-792 ... exhibited dramatically lower appearance of SUMOylated ACE2’ – this should be corrected to ‘lower co-localization of SUMO2/3 and ACE2’ since the microscopy assay does only allow for conclusions about localization of ACE2 and SUMO, not about actual SUMO-conjugation to ACE2.

Response: We have corrected our text as the reviewer pointed out.

13. I think the authors should make the full gene list from RNA-Seq experiments available to enable the reader to recapitulate the presented GO and GSEA analyses (data used for Fig. 2 and Fig. 5)

Response: We have provided the full gene lists of our RNA-Seq experiments in the supplementary Table 7 and 8.

14. Fig 2: I suggest to add experiments to demonstrate that the inhibitor treatment used in this experiment does not affect viability of the Calu-3 cells, since this could also lead to impaired virus replication.

Response: We thank the reviewer for this suggestion. We assessed the cytotoxicity of SUMOylation inhibitors using a lactate dehydrogenase (LDH) assay, observing that only the high concentration GA (100 μ M) and TAK1-981 (50 to 100 μ M) had slight influence on the viability of Calu-3 cells (**New Fig. 6**, related to Supplementary Fig. 2a). The concentration of 2-D08 and GA we employed in Fig. 2 had no significant effect on cell viability. We have added this results in our revised manuscript.

New Figure 6. Calu-3 cell viability measured by the LDH assay when incubated with different concentrations of SUMOylation inhibitors for 24 hr. Data represented mean \pm SEM of triplicate experiments. * $p < 0.05$; NS, not significant (two-tailed Student's t -test).

15. For Figure 2 and 5, the authors state in the figure legend that these are microarray data. However, in the main text and the method part they talk about RNA seq analysis. Please clarify.

Response: We apologize for the wrong statement. All the transcriptome data were coming from RNA-seq analysis and we have corrected these mistakes in the revised manuscript.

16. The siRNA transfection experiments in Fig. 3g,h,i as well as in Fig. 5e,f,g,h and Ext.Fig 5g are only done with one siRNA per gene. It would useful to repeat the assays with a second siRNA, to exclude that the observed phenotypes are caused by potential off-target effects of the siRNA.

Response: We employed two siRNAs to repeat the experiments in Fig. 3g,h,i and Fig. 5e (**New Fig. 7a to 7d**, related to Fig. 3g,h,j and Fig. 5f in the revised manuscript).

For the results of RNA-seq analysis in Fig. 5f,g,h, we added another siRNA targeting *TOLLIP* for qPCR validation and obtained a similar result, which indicating that our RNA-seq analysis is reliable (New Fig. 7e, related to Supplementary Fig. 5j). Collectively, these results can exclude potential off-target effects of the siRNA and we have added the sequence of siRNA we used in the Supplementary Table 4.

New Figure 7. (a–c) Calu-3 cells were transfected with scramble siRNA as well as *SENP3*-specific (a), *PIAS4*-specific (b), or *TOLLIP*-specific (c) siRNAs, followed by SARS-CoV-2 (MOI = 0.5) infection for indicated time points. The protein extracts were harvested for immunoblot analysis. (d) 293T cells transfected with scramble or *SENP3*-specific siRNAs were then transfected with plasmids encoding Flag-ACE2 and HA-SUMO3. The protein extracts were harvested for immunoprecipitation and immunoblot analysis. (e) Calu-3 cells were transfected with scramble or *TOLLIP*-specific siRNAs treated with SARS-CoV-2 (MOI = 0.5) for indicated time points. Relative expression levels of selected genes were measured by qPCR. Data in (j) is expressed as means \pm SEM of at least three independent experiments. *** p <

0.001 (two-tailed Student's *t*-test). For data in (a–d), similar results were obtained by three independent biological experiments.

17. line 192-196 / Fig. 4a: It looks like ACE2 is upregulated in the presence of the inhibitors and does not further increase by overexpression of PIAS4, rather than the PIAS4-mediated upregulation being blocked as stated by the authors. Adding a summary quantification analysis of the western blot signals to visualize the results of the three independent experiments could help the reader to judge the data.

Response: We have reanalyzed our results of PIAS4-blocked autophagic degradation of ACE2 and displayed the quantification results of our western blotting data from at least three independent experiments (**New Figure 8**, related to Supplementary Fig. 4a).

New Figure 8. (a–b) 293T cells were transfected with plasmid encoding Flag-ACE2 together with Myc-PIAS4 vector treated with MG132 (10 μM), 3-MA (10 mM), or Baf A1 (0.2 μM) for 6 hr. The cell lysates were analyzed by immunoblot. (c) The quantification of ACE2 expression from Fig. 4a, New Fig. 7a and 7b. Data in (c) were expressed as means ± SEM of three independent experiments. ****p* < 0.001; NS, not significant (two-tailed Student's *t*-test).

18. In general, it would be useful to add a quantification analysis of western blot signals obtained in independent experiments for all western blots where the differences are difficult to judge from the picture of only one membrane (e.g. Fig. 4b,d,e; Ext.Fig. 5e,f; Fig.6a)

Response: We have reanalyzed our results of at least three independent experiments, and displayed the quantification results of our western blotting data in Fig. 4 of the

revised edition (**New Figure 9**, related to Supplementary Fig. 5b,d,e).

New Figure 9. (a and b) Lysates of Calu-3 cells transfected with scramble or *PIAS4*-specific siRNA, followed by rapamycin (250 nM) treatment for 24 hr, were harvested for immunoblot analysis. (c) The quantification of ACE2 expression from

Fig. 4b, New Fig. 8a and 8b. **(d and e)** 293T cells transfected with scramble siRNA along with *RBICC1*- or *ATG13*-specific siRNAs were transfected with plasmids encoding Flag-ACE2 and Myc-PIAS4, the lysates were analyzed by immunoblot. **(f)** The quantification of ACE2 expression from Fig. 4d, New Fig. 8d and 8e. **(g and h)** Wild-type (WT), *BECN1* and *ATG5* knockout (KO) 293T cells were transfected with plasmids encoding Flag-ACE2 and Myc-PIAS4, the lysates were analyzed by immunoblot. **(i)** The quantification of ACE2 expression from Fig. 4e, New Fig. 8g and 8h. **(j and k)** Immunoblot analysis of Calu-3 cells transfected with scramble or *TOLLIP*-specific siRNAs for 8 hr, followed by *PIAS4*-specific siRNA transfection. **(l)** The quantification of ACE2 expression from Extended Data Fig. 5e, New Fig. 8d and 8e. **(m and n)** Immunoblot analysis of Calu-3 cells transfected with scramble or *TOLLIP*-specific siRNAs for 8 hr, followed by *SENP3*-specific siRNA transfection. **(o)** The quantification of ACE2 expression from Extended Data Fig. 5e, New Fig. 8d and 8e. **(p and q)** 293T cells were transfected with plasmids encoding Flag-ACE2 and HA-TOLLIP, and treated with indicated concentrations of 2-D08 for 24 hr. The cells cultured with Baf A1 (0.2 μ M) for 6 hr were immunoprecipitated with anti-Flag and immunoblotted with anti-HA. **(r)** The quantification of ACE2 expression from Fig. 6a, New Fig. 8d and 8e. Data in **(c, f, i, l, o and r)** were expressed as means \pm SEM of three independent experiments. * $p < 0.05$, ** $p < 0.01$, *** $p < 0.001$; NS, not significant (two-tailed Student's *t*-test).

19. line 198: It should be explained how Rapamycin and EBSS work in the cell.

Response: Rapamycin binds to FKBP1A/FKBP12 and inhibits mTORC1, thus inducing autophagy, while Earle's balanced salts solution (EBSS) is a saline solution with physiological pH which is often used to induce starvation for autophagy activation¹³. We have added the working mechanisms of rapamycin and EBSS in the manuscript.

20. Fig. 4i: Could you explain why the fractionation is performed in the presence of 3-MA? How does this affect the results?

Response: We are sorry for causing the misunderstanding from the reviewer. 3-MA is a PtdIns3K inhibitor that effectively blocks an early stage of autophagy by inhibiting the class III PtdIns3K, VPS34¹³. We used 3-MA to rescue the ML-792-induced autophagic degradation of ACE2. If we did not add 3-MA, the protein abundance of ACE2 will be remarkably decreased in the ML-792-treated group, then the samples are not suitable for further membrane protein extraction assay. We treated the cells with 3-MA to harvest the samples with equal total protein levels of ACE2 for the comparison of the distribution of ACE2 in different fractionations.

21. *Fig. 4h: What does 'Group 1', 'Group 2', 'Group 3' mean?*

Response: We are sorry for causing this confusion. In fact, they are images of three independent repeats. We have changed the labels in the revised figures.

22. *Fig 5b: Showing in addition a higher magnification (zoom in) of the microscopy pictures would help the reader to judge the images.*

Response: We have added a higher magnification (zoom in) of the microscopy pictures as the reviewer suggested.

23. *Ext. Fig 4g: A description of the ACE2 Δ CT+TM mutant is missing. It is not clear, which parts of the protein have been deleted in this mutant. It would also be useful to cite respective literature where this mutant was described as I believe it is supposed to act as a control in this assay.*

Response: We are sorry for the ignorance to introduce the domain deletion of ACE2. The C terminal collectrin-like-domain (CLD, aa 615–805) of ACE2 consists of a single transmembrane (TM) helix (aa 740–762) and an intracellular tail (CT, aa 763–805)^{14, 15}. The ACE2 Δ CT+TM mutant was used as a negative control as the reviewer thought and we have added the description of CT+TM in the manuscript.

24. *line 306/307: 'K48-linked ubiquitination of ACE2 was enhanced in SENP3-deficient cells' – 'ubiquitination' should be replaced by 'co-localization' since*

this is an IF assay that does not allow for conclusion about covalent ubiquitination.

Response: We are revised our manuscript as the reviewer suggested.

25. *The following experimental details are missing in the methods section:*

- o Infection assay using Spike-pseudotyped VLPs*
- o SARS-CoV-2 virus stock production*
- o Description of many of the used expression plasmids*

Response: We have added the experimental details in the methods section.

Response to the comments of Reviewer #2

1. Although the authors managed to generate some in vivo evidence that is indeed supportive for the role of SUMOylation on SARS-CoV-2 infection, this part of the manuscript does not provide sophisticated evidence on the proposed link of SUMOylation, ACE2 abundance and SARS-CoV-2 infection. The manuscript would strongly improve if conclusive data on SUMOylation-dependent ACE2 expression locally in infected cells could be obtained. The proposed mouse model that relies on Ad5- mediated hACE2 expression and the systemic application of SUMOylation inhibitors is questionable regarding its clinical relevance, but also regarding its suitability in the context of the paper. Are both conjugation processes, SUMO+ubi, as conserved as needed to conclude on ACE-2 abundance? Is SUMOylation indeed affected by the respective inhibitors, how is ACE-2 in AEC controlled by these aspects, is indeed the viral entry affected or the replication process affected by alternative SUMOylation-dependent processes. Coronaviruses, at least partially, utilize the autophagic pathway for replication. In addition the demonstration/elucidation of the respective pathways in lung tissue, the proposed effects on the SARS-CoV-2 triggered immune response deserve further detailed demonstration, e.g. of lung pathology, CT scans, BAL etc. to conclude part is elegant, but less well documented in terms of viral-infection triggered immune responses.

Response: We thank this reviewer for this comment. Ad5-hACE2-transduced mice can provide broad and immediate utility to investigate COVID-19 pathogenesis and to

evaluate new therapies and vaccines¹⁶. Our study indicated that SUMOylation affected the protein stability of ACE2, thus this murine model is suitable for our investigation as the expression of hACE2 is not disturbed by transcription regulation in Ad5-hACE2-transduced mice. We agree with the reviewer that the systemic application of SUMOylation inhibitors may have some potential irrelevance of mouse model, perhaps local drug delivery should be developed in the future clinical research. To investigate the role of SUMOylation-dependent ACE2 stabilization in infected cells, we transfected 293T cells with plasmids encoding Flag-ACE2 and K187R mutant form of ACE2, in which the ability to conjugate with SUMO is abrogated, and infected the cells with pseudotyped SARS-CoV-2 in the absence or presence of SUMOylation inhibitor. As 2-D08 prompts the degradation of WT ACE2, but not K187R ACE2 mutant, we observed that the suppression of SARS-CoV-2 infection induced by 2-D08 treatment was abolished in K187R ACE2-expressing cells (**New Fig. 10a**, related to Supplementary Fig. 6h). This result suggested that SUMOylation facilitates SARS-CoV-2 infection by promoting the protein stability of ACE2.

To resolve the reviewer's concern on the function of SUMOylation in modulating ACE2 abundance, we observed the SUMOylation of ACE2 was significantly decreased with the treatment of SUMOylation inhibitors (**New Fig. 10b**, related to Fig. 2b in the revised manuscript). As the reviewer suggested, we isolated alveolar epithelial cells (AECs) from human, and treated the AECs with SUMOylation inhibitors, finding that the SUMOylation and protein stabilization of ACE2 were largely decreased, while the K48-linked ubiquitination of ACE2 was significantly increased (**New Fig. 10c**, related to Fig. 2d in the revised manuscript). The results demonstrated that ACE2 in AECs was also controlled by SUMO and Ub system. As SARS-CoV-2 S pseudotyped virus does not contain nucleic acid and cannot replicate in host cells, it can be a good model for study of viral entry. Our further results revealed that the infection of pseudotyped SARS-CoV-2 was largely abrogated by SUMOylation inhibitors in AECs (**New Fig. 10d**, related to Fig. 2e in the revised manuscript). Together, these results suggested that SUMOylation and ubiquitination control the protein stability of ACE2, which influence the SARS-CoV-2 infection and

provide potential therapeutic targets for SARS-CoV-2.

We agree with the reviewer that coronaviruses may partially utilize autophagy pathway for replication. However, several recent researches indicated that SARS-CoV-2 infection suppresses autophagy pathway. ORF3a of SARS-CoV-2 blocks HOPS complex-mediated assembly of the SNARE complex required for autolysosome formation¹⁷. SARS-CoV-2 infection activates negative regulators of autophagy, reduced autophagy-enhancing proteins, and limited autophagic flux by diminishing Beclin-1/ATG14-dependent autophagosome-lysosome fusion¹⁸. Moreover, SARS-CoV-2 NSP6 targets hybrid pre-autophagosomal structure (HyPAS) apparatus to inhibit prophagophore formation¹⁹. Therefore, the interplay between autophagy and SARS-CoV-2 infection is quite complicated. The SUMOylation-mediated ACE2 stabilization was not caused by affecting the global macroautophagy (**New Fig. 12a**, related to Supplementary Fig. 4j). Moreover, the basal levels of ACE2 were much lower in *PIAS4*-deficient cells, and knockdown of *SENP3/TOLLIP* led to the accumulation of ACE2 even without SARS-CoV-2 infection (**New Fig. 7a to 7c**, related to Fig. 3g,j and 5f in the revised manuscript). These results suggested that selective autophagy may influence the basal protein level of ACE2 through SUMOylation.

Studying the common dynamics of immune response changes by analysis of the computed tomography (CT), lung pathology, and bronchoalveolar lavage (BAL) fluid, which can help understand the pathogenesis of COVID-19. In our manuscript, we demonstrated that SUMOylation of ACE2 contributes to the stabilization of ACE2, which may play an important role in SARS-CoV-2 infection. To further investigate SARS-CoV-2-triggered immune response as the reviewer suggested, we harvested the bronchoalveolar lavage (BAL) fluid from the infected Ad5-hACE2-transduced mice, and observed infiltrating neutrophils, granulocytes and inflammatory monocytes were elevated in BAL, while they were largely decreased by SUMOylation inhibitor treatment (**New Fig. 10e**). Taken together, our findings suggested that SARS-CoV-2 infection was shut down by SUMOylation regulation of ACE2 through selective autophagy. Our study mainly focuses on the SARS-CoV-2 infection modulated by

ACE2 SUMOylation, the regulatory roles of global macroautophagy as well as SUMOylation of host immune factors and viral proteins in contributing to SARS-CoV-2-triggered immune responses merit further study in the future.

New Figure 10. (a) Luciferase activity of 293T cells transfected with plasmid expressing WT or K187R mutant form of ACE2, along with SARS-CoV-2 S pseudotyped virus infection for 48 hr in the absence or presence of 2-D08 (200 μ M). (b) 293T cells transfected with plasmids expressing Flag-ACE2 and HA-SUMO3 were treated with 2-D08 (200 μ M), ML-792 (10 μ M), GA (5 μ M), or TAK-981 (5 μ M). The lysates were harvested for immunoblot analysis with indicated antibodies. (c) Human alveolar epithelial cells treated with ML-792 (10 μ M) or GA (5 μ M) were

subjected to immunoprecipitation with anti-ACE2 and immunoblot analysis with indicated antibodies. **(c)** Luciferase activity of human alveolar epithelial cells treated with ML-792 (10 μ M) or GA (5 μ M), along with SARS-CoV-2 S pseudotyped virus infection for 24 hr. **(d)** Ad5-hACE2-transduced BALB/c mice were intranasally infected with 1×10^5 PFU of SARS-CoV-2 in 50 mL of DMEM. Bronchoalveolar lavage was collected, cells isolated, and stained via flow cytometry. Data is expressed as means \pm SD of 7 mice. Two-way ANOVA was performed to calculate the p values. *** $p < 0.001$. Data in **(a and d)** are expressed as means \pm SEM of three independent experiments. ** $p < 0.01$ (two-tailed Student's t -test). For data in **(b and c)**, similar results were obtained by three independent biological experiments.

2. More data is required on the role of SUMOylation on SARS-CoV-2 replication. Fig 5f is a linear or Log10 scale? There is a lot of indirect evidence for the proposed mechanisms that affect SARS-CoV-2 entry. The authors should explain in detail which experiments precisely document virus entry as opposed to altered replication and (thereby influenced) immune/cellular responses.

Response: Fig. 5f is a linear scale. We employed the SARS-CoV-2 S pseudotyped virus and performed more experiments in *PIAS4* and *SENP3* deficient cells, and observed that *PIAS4* knockdown decreased the amount of cellular pseudotyped SARS-CoV-2, while *SENP3* depletion increased the pseudotyped SARS-CoV-2 infection (**New Fig. 1a and 1b**, related to Fig. 3f,i in the revised manuscript). We hope the added new experiments can strengthen our proposed mechanism of SARS-CoV-2.

3. Several typos and grammar issues need to be addressed.

Response: We have revised our manuscript as the reviewer suggested.

4. Since many different inhibitors are used, it is recommended to provide a table, listing the compounds and their respective targets.

Response: We have added a table to list the compounds and their respective targets in Supplementary Table 2.

Response to the comments of Reviewer #3

1. The first important point is the effects and concentrations of Bafilomycin (200 nM, 6 hrs) and 3-MA (10 mM 6 hrs) and Rapa (250 nM). These concentration of inhibitors could be extremely toxic for some cell lines. It is important the respected authors provide evidence about the viability of cells in all conditions.

Response: We thank the reviewer for this suggestion and checked the viability of cells in all inhibitor treatment conditions. We assessed the cytotoxicity of bafilomycin, 3-MA and rapamycin using a lactate dehydrogenase (LDH) assay, observing that treatment with these chemicals had no significant influence on the viability of both 293T and Calu-3 cells (**New Fig. 11a to 11d**). Only the long time (24 hr) of rapamycin (250 nM) treatment had slight influence on the viability of Calu-3 cells (**New Fig. 11e**).

New Figure 11. (a–c) 293T cell viability measured by the LDH assay when incubated with 3-MA (200 nM, 6 hr) (a), Baf A1 (10 mM, 6hr) (b), and rapamycin (250 nM, 12 hr) (c). (d) Calu-3 cell viability measured by the MTT assay when incubated with rapamycin (250 nM) for 12 hr. (e) The cell viability of hNEPCs with rapamycin treatment for indicated time points was measured by the MTT assay. Data in (a–e) represented mean \pm SEM of triplicate experiments. * $p < 0.05$; NS, not significant. (two-tailed Student's t -test).

2. It is strongly recommended that the respected authors provide blots in each conditions for LC3 lipidation, p62 degradation, Beclin-1 and Atg5-12 conjugations.

Response: We thank the reviewer for this constructive suggestion. Actually, in most conditions in Fig. 3, we detected the degradation of ACE2 without rapamycin or EBSS treatment. We found that the SUMOylation inhibitors could not significantly

affect the LC3 lipidation, p62 degradation, Beclin-1 abundance and Atg5-12 conjugations upon different conditions chosen in our system (**New Fig. 12a**, related to Supplementary Fig. 4j). Under many autophagy-induced conditions, the protein levels of Beclin-1 and Atg5-12 conjugation remain largely unchanged. Immunoblot analysis of the protein level of Beclin-1 and Atg5-12 conjugation may be not a sensitive standard for monitoring autophagy¹³. Moreover, we analyzed LC3 lipidation, p62 degradation, Beclin-1 and Atg5-12 conjugations to study the autophagic degradation of ACE2 K187R mutant. The degradation of ubiquitination null mutant form of ACE2 was significantly abrogated under the macroautophagy activation conditions (**New Fig. 12b**). Together, these results indicated that the selective autophagic pathway contributes to SUMOylation mediated ACE2 stabilization.

New Figure 12. (a) Immunoblot analysis of Calu-3 cells treated with 2-D08 (200 μM), ML-792 (10 μM), GA (5 μM), TAK-981 (5 μM), or rapamycin (250 nM). (b) Immunoblot analysis of 293T cells transfected with Flag-ACE2 (WT or K187R) and treated with rapamycin (250 nM) for indicated time points. For data in (a and b), similar results were obtained by three independent biological experiments.

3. Based on the data that has been presented, I believe beyond selective autophagy, we may have effects of autophagy in the intracellular trafficking and its effect of ACE2. I recommend the respected authors provide evidence for using different concentration of Baf-A1 (0.1, 1, 2.5, 10 nM) and co-treat them with the SARS-CoV-2 during all infection time and look at ACE2 expression and also localize ACE-2 with early and late endosomes, and early late exosomes in different Baf-A1 concentrations. It may give us a much better understanding of the role of autophagy in ACE-2 life

cycle in the infected cells.

Response: We agree with the reviewer that autophagy may affect the intracellular trafficking and effect of ACE2. We firstly examined the protein abundance of ACE2 with Baf A1 (0.1, 1, 2.5, 10 nM) treatment together with SARS-CoV-2 infection, observing that SARS-CoV-2 infection-induced ACE2 down-regulation could be rescued by even the low concentration of Baf A1 treatment (**New Fig. 13a**). As Baf A1 is a vacuolar H⁺-ATPase inhibitor to prevent re-acidification of synaptic vesicles following exocytosis and block fusion of autophagosomes and lysosomes¹³. These results indicated the down-regulation of ACE2 protein can be caused by both endocytosis and autophagic degradation. However, our previous results revealed that the SUMOylation inhibitor treatment-induced ACE2 degradation cannot be rescued by endocytosis inhibitors (**Fig. 4f** and **4g**), suggesting SUMOylation mainly maintains the stabilization of ACE2 through selective autophagy pathway.

We also investigated the localization of ACE2 under two chosen concentration of Baf A1 treated conditions. We found that ACE2 mainly co-localized with EEA1 (an early endosome marker) at 24 hr after SARS-CoV-2 infection, and most ACE2 distributed together with Rab7 (a late endosome marker) at 48 hr post SARS-CoV-2 treatment (**New Fig. 13b, 13c, 13e** and **13f**). These results suggested that ACE2 may co-localize with early endosome at early stage of infection, but co-localize with late endosome at late stage of SARS-CoV-2 infection. Moreover, we detected there was no co-localization between ACE2 and CD63 (an early exosome marker) at early stage of infection, but a slight co-localization at late stage of SARS-CoV-2 infection (**New Fig. 13d** and **13g**). Our results showed the life cycle of ACE2 in the infected cells. As the basal levels of ACE2 were much lower in *PIAS4*-deficient cells, and knockdown of *SEN3/TOLLIP* led to the accumulation of ACE2 even without SARS-CoV-2 infection (**New Fig. 7a** to **7c**, related to Fig. 3g,j and 5f in the revised manuscript), we suggested that selective autophagy may influence the basal protein level of ACE2.

New Figure 13. (a) Calu-3 cells treated with Baf A1 (0.1, 1, 2.5, 10 nM) were infected with SARS-CoV-2 (MOI = 0.5) for indicated time points. The protein extracts were harvested for immunoblot analysis. (b–d) Confocal microscopy of ACE2-expressing A549 cells treated with Baf A1 (1 or 10 nM) were infected with SARS-CoV-2 (MOI = 0.5) for indicated time points, followed by labeling of ACE2 (green) and EEA1 (red) (b), Rab7 (red) (c), and CD63 (red) (d) with specific antibodies. Scale bar, 20 μ m. (e–f) Quantitative analysis of the similar samples as (b) to (d). Data in (e–f) are expressed as means \pm SD of 10 cells per sample. * p < 0.05, ** p < 0.01, *** p < 0.001; NS, not significant (two-tailed Student's t -test). For data in (a), similar results were obtained by three independent biological experiments.

4. We all know the secretory autophagy is playing a substantial role in cellular immune response of the infected cells. It is important that the respected authors provide evidence about the role of secretory autophagy in regulation of cytokine/chemokine profile of the infected cells using 2 low concentration of Baf-A1. It may have effect in ACE2 recycling process.

Response: We thank the reviewer for this great idea, and we have analyzed the cytokine/chemokine profile of SARS-CoV-2 infected-Calu-3 cells with two low concentrations of Baf A1 treatment. We found that the release of IL-1 β , IL6 and LIF, the previously reported secreted entities regulated by autophagy-dependent secretion²⁰, was largely abrogated by treatment with Baf A1. However, the secretion of a series of other cytokines and chemokines which are uncertain whether can be release through secretory autophagy were also down-regulated by Baf A1 treatment (**New Fig. 14**). Inhibition of the endosomal acidification may suppress SARS-CoV-2 replication, thus leading to the weaker cytokine/chemokine expression, which was consistent with previous reports^{21,22}. However, decreased secretory autophagy may contribute to the ACE2 recycling process to inhibit SARS-CoV-2 infection. We believe secretory autophagy in SARS-CoV-2 infection can be a very interesting project for the further study.

New Figure 14. Heat maps of cytokine levels, as measured by multiplex platform, in SARS-CoV-2-infected Calu-3 cells together with Baf A1 (0.1 or 25 nM) treatment. For each cytokine, the fold change was calculated as compared with mock-infected cells and the \log_2 [fold change] was plotted in the corresponding heat map.

5. *The data for the Cav-1 is very interesting. It directs us the potential role of SARS-CoV-2 lipophagy and indirect effect of lipid raft. It is recommended the respected authors provide evidence about the potential role of SARS-CoV-2 infection in lipophagy and also use lipidomics to measure different contents of lipids like cholesterol and ceramides that are important for lipid raft and cytosolic membranes.*

Response: We thank the reviewer for his/her interest in the lipophagy and we have performed additional experiments as the reviewer suggested. Lipid droplets (LDs), are lipid-rich cellular organelles that regulate lipid homeostasis and metabolism. They serve as a store for cholesterol esters and triacylglycerols (TAGs) utilized for membrane formation and maintenance but also supply essential lipids to produce signaling molecules and metabolic energy²³. The level of intracellular lipophagy was visualized by the colocalization of lipid droplets (BODIPY 493/503 staining) and lysosomes (lysosome marker, LAMP2). We observed that SARS-CoV-2 infection dramatically enhanced the lipophagy (**New Fig. 15**), and strongly believed that

SARS-CoV-2 induced lipophagy might be another good research field.

When comes to the lipidomics, several groups had already conducted elegant studies. By monitoring a cohort of COVID-19 patients' plasma lipidome over the disease course, researchers identified triacylglycerol (TG) as the dominant lipid class present in SARS-CoV-2-induced metabolic dysregulation. Virus manipulates host glycerophospholipid metabolism through upregulation of TG synthesis and LD formation. Specifically, viral NP trans-activates both *DGAT1* and *DGAT2* mRNA expression binding to their promoter regions, resulting in LD accumulation. Both DGATs and the other LD surface protein ADRP are required for SARS-CoV-2 replication²⁴. COVID-19 patients are also characterized by an increase of serum TAG, TAG-VLDL, TAG-IDL (IDL, intermediate density lipoproteins), TAG-LDL (LDL, low-density lipoproteins) and TAG-HDL (HDL, high density lipoproteins), while a decreased of total cholesterol (TC). All these examinations imply a remodeling of the lipoprotein particle phenotype in COVID-19 patients, and thus of a great part of circulating lipids²⁵.

Cholesterol may be a predictive marker of poor outcome for COVID-19 infection and COVID-19 patients on statins had better outcome²⁶. A decreased rate of SARS-CoV-2 infection under statin treatment, possibly due to reduced membrane cholesterol impacting ACE2-viral spike protein interactions²⁷. Pseudoviral SARS-CoV-2 induces the acid sphingomyelinase/ceramide system, and inhibition of acid sphingomyelinase prevents cellular infection with SARS-CoV-2²⁸. As ACE2 clusters in ceramide-enriched membrane domains upon infection, inhibition of acid sphingomyelinase by ambroxol prevents SARS-CoV-2 entry into epithelial cells²⁹. Altogether, lipophagy and lipidomics under SARS-CoV-2 infected conditions are very interesting research fields and we have added part of this content in our discussion.

New Figure 15. (a) Confocal microscopy of ACE2-expressing A549 cells infected with SARS-CoV-2 (MOI = 0.5) for indicated time points, followed by labeling of LAMP2 (red) with specific antibody and lipid droplets (green) with BODIPY, respectively. Scale bar, 20 μ m. (b) Quantitative analysis of the similar samples as (a). Data in (b) are expressed as means \pm SD of 10 cells per sample. *** $p < 0.001$ (two-tailed Student's t -test).

References

1. Lei CH, *et al.* Neutralization of SARS-CoV-2 spike pseudotyped virus by recombinant ACE2-Ig. *Nature Communications* **11**, 1-5 (2020).
2. Bewley KR, *et al.* Quantification of SARS-CoV-2 neutralizing antibody by wild-type plaque reduction neutralization, microneutralization and pseudotyped virus neutralization assays. *Nature Protocols* **16**, 3114-3140 (2021).
3. Raman N, Nayak A, Muller S. mTOR signaling regulates nucleolar targeting of the SUMO-specific isopeptidase SENP3. *Molecular and Cellular Biology* **34**, 4474-4484 (2014).
4. Wei B, *et al.* Mitotic phosphorylation of SENP3 regulates deSUMOylation of chromosome-associated proteins and chromosome stability. *Cancer Research* **78**, 2171-2178 (2018).
5. Wang Y, *et al.* P53 suppresses SENP3 phosphorylation to mediate G2 checkpoint. *Cell discovery* **6**, 1-9 (2020).
6. Appelberg S, *et al.* Dysregulation in Akt/mTOR/HIF-1 signaling identified by

proteo-transcriptomics of SARS-CoV-2 infected cells. *Emerging Microbes & Infections* **9**, 1748-1760 (2020).

7. Bock JO, Ortea I. Re-analysis of SARS-CoV-2-infected host cell proteomics time-course data by impact pathway analysis and network analysis: a potential link with inflammatory response. *Aging-Us* **12**, 11277-11286 (2020).

8. Zhang J, *et al.* AMP-activated protein kinase phosphorylation of angiotensin-converting enzyme 2 in endothelium mitigates pulmonary hypertension. *American Journal of Respiratory and Critical Care Medicine* **198**, 509-520 (2018).

9. Su S, *et al.* Lenalidomide downregulates ACE2 protein abundance to alleviate infection by SARS-CoV-2 spike protein conditioned pseudoviruses. *Signal Transduction and Targeted Therapy* **6**, 1664-1667 (2021).

10. Tu WJ, *et al.* Targeting novel LSD1-dependent ACE2 demethylation domains inhibits SARS-CoV-2 replication. *Cell Discovery* **7**, 1-25 (2021).

11. Donoghue M, *et al.* A novel angiotensin-converting enzyme-related carboxypeptidase (ACE2) converts angiotensin I to angiotensin 1-9. *Circulation Research* **87**, 1-9 (2000).

12. Rutz C, Klein W, Schulein R. N-terminal signal peptides of G protein-coupled receptors: significance for receptor biosynthesis, trafficking, and signal Transduction. In: *Trafficking of Gpcrs* (ed Wu G) (2015).

13. Klionsky DJ, *et al.* Guidelines for the use and interpretation of assays for monitoring autophagy (4th edition). *Autophagy* **17**, 1-382 (2021).

14. Gross LZF, Sacerdoti M, Piiper A, Zeuzem S, Leroux AE, Biondi RM. ACE2, the receptor that enables infection by SARS-CoV-2: biochemistry, structure, allostery and evaluation of the potential development of ACE2 modulators. *Chemmedchem* **15**, 1682-1690 (2020).

15. Sharma RK, *et al.* ACE2 (angiotensin-converting enzyme 2) in cardiopulmonary diseases ramifications for the control of SARS-CoV-2. *Hypertension* **76**, 651-661 (2020).

16. Sun J, *et al.* Generation of a broadly useful model for COVID-19 pathogenesis, vaccination, and treatment. *Cell* **182**, 734-743 (2020).

17. Miao GY, *et al.* ORF3a of the COVID-19 virus SARS-CoV-2 blocks HOPS complex-mediated assembly of the SNARE complex required for autolysosome formation. *Developmental Cell* **56**, 427-442 (2021).
18. Gassen NC, *et al.* SARS-CoV-2-mediated dysregulation of metabolism and autophagy uncovers host-targeting antivirals. *Nature Communications* **12**, 1-15 (2021).
19. Kumar S, *et al.* Mammalian hybrid pre-autophagosomal structure HyPAS generates autophagosomes. *Cell* **184**, 5950-5969 (2021).
20. New J, Thomas SM. Autophagy-dependent secretion: mechanism, factors secreted, and disease implications. *Autophagy* **15**, 1682-1693 (2019).
21. Shang C, *et al.* Inhibitors of endosomal acidification suppress SARS-CoV-2 replication and relieve viral pneumonia in hACE2 transgenic mice. *Virology Journal* **18**, (2021).
22. Prabhakara C, *et al.* Strategies to target SARS-CoV-2 entry and infection using dual mechanisms of inhibition by acidification inhibitors. *Plos Pathogens* **17**, (2021).
23. Casari I, Manfredi M, Metharom P, Falasca M. Dissecting lipid metabolism alterations in SARS-CoV-2. *Progress in Lipid Research* **82**, 1-17 (2021).
24. Yuan SF, *et al.* SARS-CoV-2 exploits host DGAT and ADRP for efficient replication. *Cell Discovery* **7**, 1-13 (2021).
25. Bruzzone C, *et al.* SARS-CoV-2 Infection Dysregulates the Metabolomic and Lipidomic Profiles of Serum. *Iscience* **23**, 1-25 (2020).
26. Zhang XJ, *et al.* In-hospital use of statins is associated with a reduced risk of mortality among individuals with COVID-19. *Cell Metabolism* **32**, 176-187 (2020).
27. Zapatero-Belinchon FJ, *et al.* Fluvastatin mitigates SARS-CoV-2 infection in human lung cells. *Iscience* **24**, 1-24 (2021).
28. Carpinteiro A, *et al.* Pharmacological Inhibition of Acid Sphingomyelinase Prevents Uptake of SARS-CoV-2 by Epithelial Cells. *Cell Reports Medicine* **1**, 1-17 (2020).
29. Carpinteiro A, *et al.* Inhibition of acid sphingomyelinase by ambroxol prevents

SARS-CoV-2 entry into epithelial cells. *Journal of Biological Chemistry* **296**, (2021).

REVIEWERS' COMMENTS

Reviewer #1 (Remarks to the Author):

In the revised version of the manuscript 'Suppression of ACE2 SUMOylation protects against SARS-CoV-2 infection through TOLLIP-mediated selective autophagy', Jin et al. provide sufficient experimental data to address my concerns, including additional evidence supporting the role of SUMO modification of ACE2 in SARS-CoV-2 entry using a S-pseudotyped VLP assay.

However, there are still some remaining points that should be addressed before publication of the manuscript in Nature Communications.

1. There are still unsolved concerns about the use of SUMOylation inhibitors in the mouse experiment (Fig. 2f-j). The authors did not sufficiently discuss that SUMOylation is an important regulator of transcription and the response to viral infection. I also agree with reviewer #2 who pointed out that the pathway of SUMO-regulation of ACE2 might not be conserved in mice. I recommend the authors clearly state the limitations of this experiment, including citation of respective literature about the important role of SUMO-mediated regulation of virus replication and the IFN response (as a starting point I recommend: PMID 28250012, PMID 32591223, PMID 33806893).

The authors state in their rebuttal letter 'The reason why we chosen transcriptome analysis here is that we want to provide some insights into the potential SARS-CoV-2 entry-induced immune response.' However, the read out is at 3 days post infection, which is too late to look at changes in the transcriptome caused by virus entry.

2. I think the statement in line 29/30 of the abstract ('pharmacological intervention of ACE2 SUMOylation blocks SARS-CoV-2 infection') is maybe a bit misleading, since there are no specific inhibitors of ACE2 SUMOylation available.

3. In my opinion, the data from New Figure 5 from the rebuttal letter should be included in the manuscript, e.g. in Sup. Fig. 1, since it provides an explanation for the unusual distribution of SUMO at the plasma membrane in the transfected cells, and strengthens the point that ACE2 is SUMOylated at the plasma membrane.

4. Please change the title of Supplementary Fig. 2 (Dynamic SUMOylation of ACE2 influences the entry of SARS-CoV-2). This experiment uses qPCR and RNA seq in infected mice or cells treated with

SUMO inhibitor at 24hpi – this assay is not specific for virus entry, and also not specific for ACE2 SUMOylation.

5. In line 159, the authors state: 'Altogether, these results demonstrated that suppressing ACE2 SUMOylation inhibits the SARS-CoV-2 infection.' However, in this section only global SUMO inhibitors are used, which are not specific to ACE2 SUMOylation. Thus the conclusion here should be that suppressing global SUMOylation affects ACE2 SUMOylation and inhibits SARS-CoV-2 infection.

Reviewer #2 (Remarks to the Author):

I congratulate the authors for their precise response and detailed revision of the manuscript. I agree that for many relevant findings presented in this paper substantial additional experimental evidence was generated. Moreover, I would like to thank you for your impressive review of the literature and highly recommend publication of this article. I asked the editorial team to encourage integration of the novel data sets into the final version of your manuscript.

Reviewer #3 (Remarks to the Author):

The respected authors have done fantastic job to answer my questions. Answering these comments clearly uncover some unknown aspect of SARS-COV-2 infection.

Saeid Ghavami, PhD, Associate Professor of Cancer Biology, University of Manitoba

Dear Referees,

Thank you very much for providing us a valuable opportunity to revise our paper entitled “Suppression of ACE2 SUMOylation protects against SARS-CoV-2 infection through TOLLIP-mediated selective autophagy”. A point-by-point response to the Referees’ concerns is included below. We hope that the revised manuscript meets the requirements for publication in “*Nature Communications*”, and we look forward to hearing from you.

Sincerely,

Jun Cui, Ph.D.

Professor and Dean of Department of Biochemistry

School of Life Science

Sun Yat-sen University

(The corresponding author)

Response to the comments of Reviewer #1

1. There are still unsolved concerns about the use of SUMOylation inhibitors in the mouse experiment (Fig. 2f-j). The authors did not sufficiently discuss that SUMOylation is an important regulator of transcription and the response to viral infection. I also agree with reviewer #2 who pointed out that the pathway of SUMO-regulation of ACE2 might not be conserved in mice. I recommend the authors clearly state the limitations of this experiment, including citation of respective literature about the important role of SUMO-mediated regulation of virus replication and the IFN response (as a starting point I recommend: PMID 28250012, PMID 32591223, PMID 33806893).

The authors state in their rebuttal letter 'The reason why we chosen transcriptome analysis here is that we want to provide some insights into the potential SARS-CoV-2 entry-induced immune response.' However, the read out is at 3 days post infection, which is too late to look at changes in the transcriptome caused by virus entry.

Response: We thank the reviewer for this comment. We agree with the reviewer that SUMOylation plays an important role in regulating cellular transcription and host immune response to viral infection. Our manuscript mainly focuses on the SUMOylation of ACE2 in SARS-CoV-2 infection, the SUMOylation of key proteins in immune system may also be affected by inhibitors. We have added this content and cited the recommended papers in our discussion part as the reviewer suggested. In fact, our experiments in Fig. 2f and 2g were performed in adenovirus expressing human ACE2 (Ad5-hACE2) infected mice, while the experiments in Fig. 2h-j were carried out in Calu-3 cells. We used the mouse model to study the role of SUMOylation of human ACE2 in SARS-CoV-2 *in vivo*. As mouse ACE2 (mACE2) cannot bind with the viral spike protein, SARS-CoV-2 cannot enter mouse cells. Even though the hACE2 and mACE2 are largely different, the SUMOylation system and autophagy process are quite conserved in human and mice^{1,2,3,4,5}. Our results can demonstrate the functional role of human ACE2 SUMOylation *in vivo*. Additionally, the RNA-seq analysis in Fig. 2h-j was performed after SARS-CoV-2 infection for 24 hours in Calu-3 cells. It can reflect the influence of transcriptome caused by virus entry.

2. *I think the statement in line 29/30 of the abstract ('pharmacological intervention of ACE2 SUMOylation blocks SARS-CoV-2 infection') is maybe a bit misleading, since there are no specific inhibitors of ACE2 SUMOylation available.*

Response: We are sorry for causing this misleading and have modified our abstract part as the reviewer suggested.

3. *In my opinion, the data from New Figure 5 from the rebuttal letter should be included in the manuscript, e.g. in Sup. Fig. 1, since it provides an explanation for the unusual distribution of SUMO at the plasma membrane in the transfected cells, and strengthens the point that ACE2 is SUMOylated at the plasma membrane.*

Response: We have put this figure in our supplementary information and re-integrated our data as the reviewer suggested.

4. *Please change the title of Supplementary Fig. 2 (Dynamic SUMOylation of ACE2 influences the entry of SARS-CoV-2). This experiment uses qPCR and RNA seq in infected mice or cells treated with SUMO inhibitor at 24hpi – this assay is not specific for virus entry, and also not specific for ACE2 SUMOylation.*

Response: We have changed our title of Supplementary Fig. 2 as the reviewer pointed out.

5. *In line 159, the authors state: 'Altogether, these results demonstrated that suppressing ACE2 SUMOylation inhibits the SARS-CoV-2 infection.' However, in this section only global SUMO inhibitors are used, which are not specific to ACE2 SUMOylation. Thus the conclusion here should be that suppressing global SUMOylation affects ACE2 SUMOylation and inhibits SARS-CoV-2 infection.*

Response: We thank the reviewer's constructive comment and have revised our manuscript.

Response to the comments of Reviewer #2

I congratulate the authors for their precise response and detailed revision of the manuscript. I agree that for many relevant findings presented in this paper substantial additional experimental evidence was generated. Moreover, I would like to thank you for your impressive review of the literature and highly recommend publication of this article. I asked the editorial team to encourage integration of the novel data sets into the final version of your manuscript.

Response: We would like to thank the reviewer for the positive comment regarding our manuscript and re-integrated our data as the reviewer suggested (see Supplementary Fig. 1c and Supplementary Fig. 4a–c).

Response to the comments of Reviewer #3

The respected authors have done fantastic job to answer my questions. Answering these comments clearly uncover some unknown aspect of SARS-COV-2 infection.

Response: We thank the reviewer for this comment.

References

1. Seeler JS, Dejean A. SUMO and the robustness of cancer. *Nature Reviews Cancer* **17**, 184-197 (2017).
2. Barry R, *et al.* SUMO-mediated regulation of NLRP3 modulates inflammasome activity. *Nature Communications* **9**, 1-14 (2018).
3. Hu MM, *et al.* Sumoylation promotes the stability of the DNA sensor cGAS and the adaptor STING to regulate the kinetics of response to DNA virus. *Immunity* **45**, 555-569 (2016).
4. Hansen M, Rubinsztein DC, Walker DW. Autophagy as a promoter of longevity: insights from model organisms. *Nature Reviews Molecular Cell Biology* **19**, 579-593 (2018).
5. Dikic I, Elazar Z. Mechanism and medical implications of mammalian autophagy. *Nature Reviews Molecular Cell Biology* **19**, 349-364 (2018).